# Bridging glucose metabolism and intrinsic functional organization of the human cortex
Bin Wan [1,2,3] ✉, Valentin Riedl [4,5], Gabriel Castrillon [4,5], Matthias Kirschner [3,6] & Sofie L. Valk [1,2,7] ✉

The human brain requires a continuous supply of energy to function effectively. Here, we investigated how the low-dimensional organization of intrinsic functional connectivity patterns based on resting-state functional magnetic resonance imaging relates to brain energy expenditure measured by fluorodeoxyglucose positron emission tomography. By incrementally adding more dimensions of brain organization (via functional gradients), we show that increasing amounts of variance in the map of brain energy expenditure are accounted for. Dimensions of brain organization that explained much of the variance in intrinsic brain function also accounted for a substantial share of regional variance in energy expenditure maps. This relationship was especially pronounced for maps based on the strongest connections, suggesting that weaker connections may contribute less to explaining regional energy variance. Notably, our topological model was more effective than random brain organization configurations, suggesting that brain organization may be specifically associated with energy optimization. Our results demonstrate how the spatial organization of functional connections is systematically linked to optimized energy expenditure in the human brain, providing new insights into the metabolic basis of brain function.

The human brain has rich patterns of neural activity that consume a considerable amount of metabolic energy and computational resources at rest[1–6]. However, what the regional differences in energy expenditure mean for brain function is still an open question. Previous studies have correlated cortical energy cost with regional features of the resting state functional connectome from graph analysis and found that regions with higher local connectivity, centrality, and amplitude of low frequency fluctuations consume more energy[7–12]. Although these indices provide local or network graph information about connectivity patterns, these models only partially explain the spatial pattern of energy expenditure. Another way to understand brain function is to conceptualize it as an outcome of interacting networks on a broader scale or inter-regional connectome similarity networks. Regions with similar functional connectivity (FC) profiles are spatially organised, which can be captured by low-dimensional embeddings, or gradients[13]. The gradient framework captures the local/global structure of the FC matrix and reflects both integration and segregation of functional

networks. The primary aim of this work was to study the relationships between energy consumption and organization gradients of intrinsic brain function. Moreover, both functional organization and glucose metabolism show hemispheric differences[14–21]. The classical theory of why the cortex exhibits functional asymmetry posits that it avoids costly duplication of neural circuits with the same function[22], which forms the dominant hemisphere or lateralization[23]. However, this prediction has been difficult to test because direct measurements for energy cost across the whole cortex were lacking. The second aim of our study was thus to test this theory using our energy-gradient model in different hemispheres, under the assumption that the spatial pattern of asymmetry in functional brain organization relates to the asymmetry in glucose metabolism.

Positron emission tomography (PET) and functional magnetic resonance imaging (fMRI) are suitable tools for capturing cortical energy expenditure and spatial functional organization[24–26]. In particular, the glucose analog F18 labeled deoxyglucose ($^{18}$F-FDG) is a marker of the

[1]Lise Meitner Research Group Cognitive Neurogenetics, Max Planck Institute for Human Cognitive and Brain Sciences, Leipzig, Germany. [2]Institute of Neuroscience and Medicine (INM-7: Brain and Behavior), Research Center Jülich, Jülich, Germany. [3]Department of Psychiatry, University Hospitals of Geneva, Geneva, Switzerland. [4]Department of Neuroradiology at Klinikum rechts der Isar, TUM School of Medicine and Health, Technical University of Munich, Munich, Germany. [5]Department of Neuroradiology at Uniklinikum Erlangen, Friedrich-Alexander-University Erlangen-Nuremberg, Erlangen, Germany. [6]Synapsy Center for Neuroscience and Mental Health Research, University of Geneva, Geneva, Switzerland. [7]Institute of Systems Neuroscience, Heinrich Heine University Düsseldorf, Düsseldorf, Germany. ✉e-mail: binwan@cbs.mpg.de; valk@cbs.mpg.de

intracellular glycolytic rate. Due to a lack of a 2-OH group, it cannot be further metabolized along the glycolytic pathway. PET imaging can capture the spatial distribution of accumulated F18, which then allows measurement of regional differences in the cerebral metabolic rate of glucose uptake in vivo (CMRglc)[8,27]. [18F]FDG-PET is reliable for capturing robust energy expenditure for brain activity, as the metabolic activity of brain energy expenditure is constant over time and reliable using test-retest data[26,28,29]. Resting-state fMRI captures the time-varied blood-oxygen-level-dependent signals at rest[30,31], with regions showing pairwise temporal correlation having strong connectivity, i.e., FC[32]. The resting state signal is thought to be a combination of metabolic, anatomical, and cognitive signals[33]. Regions with similar FC profiles form functional organization patterns that reflect axes of topological organization of interregional integration and segregation, i.e., gradients[13,34–36]. The first two gradients in healthy young adults describe axes that differentiate sensory from association regions and somatomotor from visual regions. They overlap with models of information processing hierarchy and microstructural variation[13,35,37], and are reproducible and reliable across subjects and sessions[36,38,39].

However, with [18F]FDG-PET, we can only obtain a map of overall energy expenditure within a region. Summarizing the functional connectome matrix to match one [18F]FDG-PET map is a challenge. To overcome this challenge, and inspired by the structural eigenmodes constraining brain function[40], we model energy expenditure using gradients of functional organization as independent variables. By evaluating the combined variance explained by each gradient, we reconstructed the regional pattern of energy expenditure, illustrating how different gradients contribute to its spatial organization. Fig. 1 illustrates the principles that regions with similar FC profiles should show similar energy expenditure. The low-dimensionality of the FC similarity matrix (i.e., affinity matrix) was then used to reconstruct the [18F]FDG-PET map.

Based on this rationale, we tested two specific hypotheses: (1) FC gradients can reconstruct regional energy expenditure patterns, with model performance improving as more gradients are included; and (2) gradient asymmetry can predict metabolic asymmetry between hemispheres. With regards to the first hypothesis, although generally only the first three gradients have been evaluated and functionally interpreted[35,41], we extended this to a more exhaustive model and observed how the model fits with the first 100 gradients. This decision was motivated by previous work that showed prediction accuracy of behavior improved by including 60 gradients or more[42]. The variance explained by these gradients may capture brain features with behavioral and physiological relevance[42]. Moreover, it will allow us to understand whether energy patterning explained by the gradient model gradually increases with the number of gradients or reaches a ceiling after a few dominant organizational principles. For the second hypothesis, we assessed asymmetry in functional gradients by training hemisphere-specific parameters. Gradient asymmetry could then be used to predict asymmetry in glucose metabolism, either directly or by training separate models for each hemisphere.

We found that combining the first 5 gradients could rival the predictive power of models based on graph topological metrics (Pearson $r$ ranging from 0.41 to 0.51), ultimately bridging global and regional features of brain organization. After 60 gradients, the variance explained converged (Pearson $r$ ranging from 0.86 to 0.88, adjusted $R$-squared ranging from 70.0 to 72.1%). Through the hemispheric modeling, we observed that the hemispheric difference of combined gradients better interpreted the glucose metabolism, rather than the hemispheric difference of single gradients (ratio = 1.14:1).

## Results
### Maps of functional organization gradients and glucose metabolism
We first downloaded the [18F]FDG-PET and fMRI images for 20 individuals from an open data source (https://openneuro.org/datasets/ds004513/versions/1.0.4). The data description and brain scanning parameters can be seen in the previous publication[8]. In short, fMRI was simultaneously

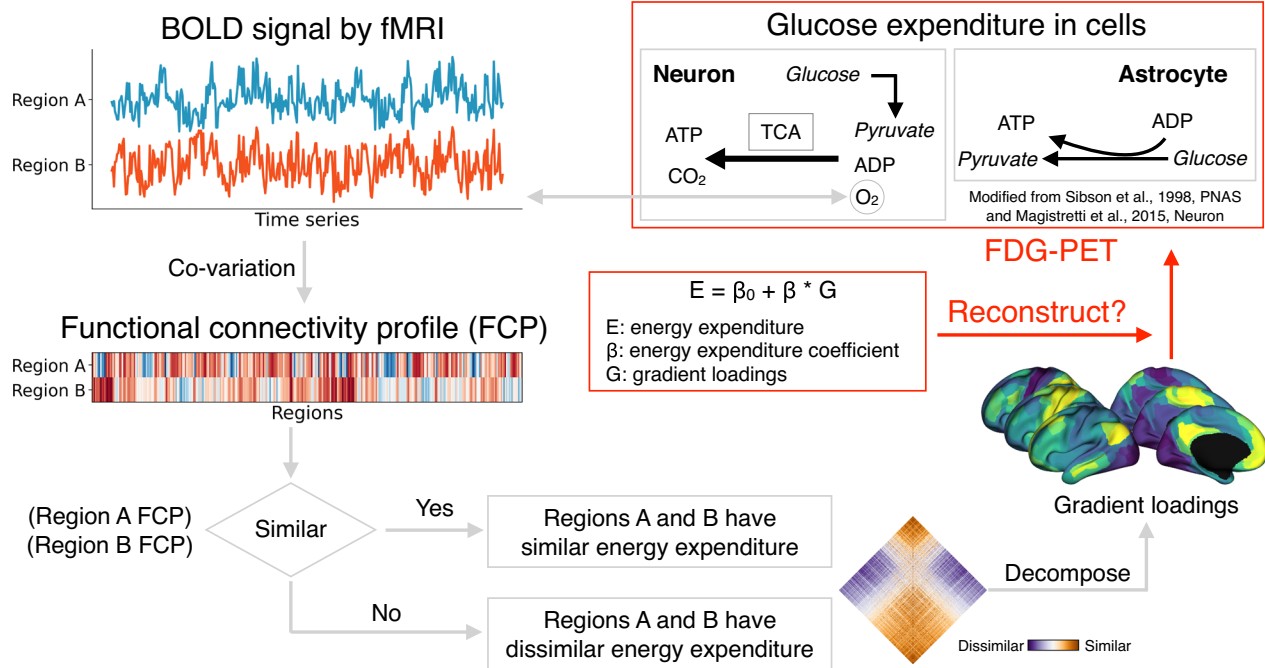

**Fig. 1 | The proposed relationship between glucose metabolism and functional organization.** Brain cells consume glucose to phosphorylate adenosine diphosphate (ADP) into adenosine triphosphate (ATP) through the tricarboxylic acid (TCA) cycle and oxidative phosphorylation. This process supports neuronal signaling underlying functional connectivity between brain regions. Neurovascular coupling is captured by the blood oxygen level–dependent (BOLD) signal, and regional functional connections are reflected by temporal synchronization of BOLD activity. Regions with similar functional connectivity profiles are hypothesized to have similar metabolic demands. We reconstructed this relationship by fitting a linear regression model (E = $\beta_0$ + $\beta$ * G), in which energy expenditure is predicted from low-dimensional functional similarity axes. Energy metabolism is measured using fluorodeoxyglucose positron emission tomography (FDG-PET), and functional activity is measured using functional magnetic resonance imaging (fMRI).

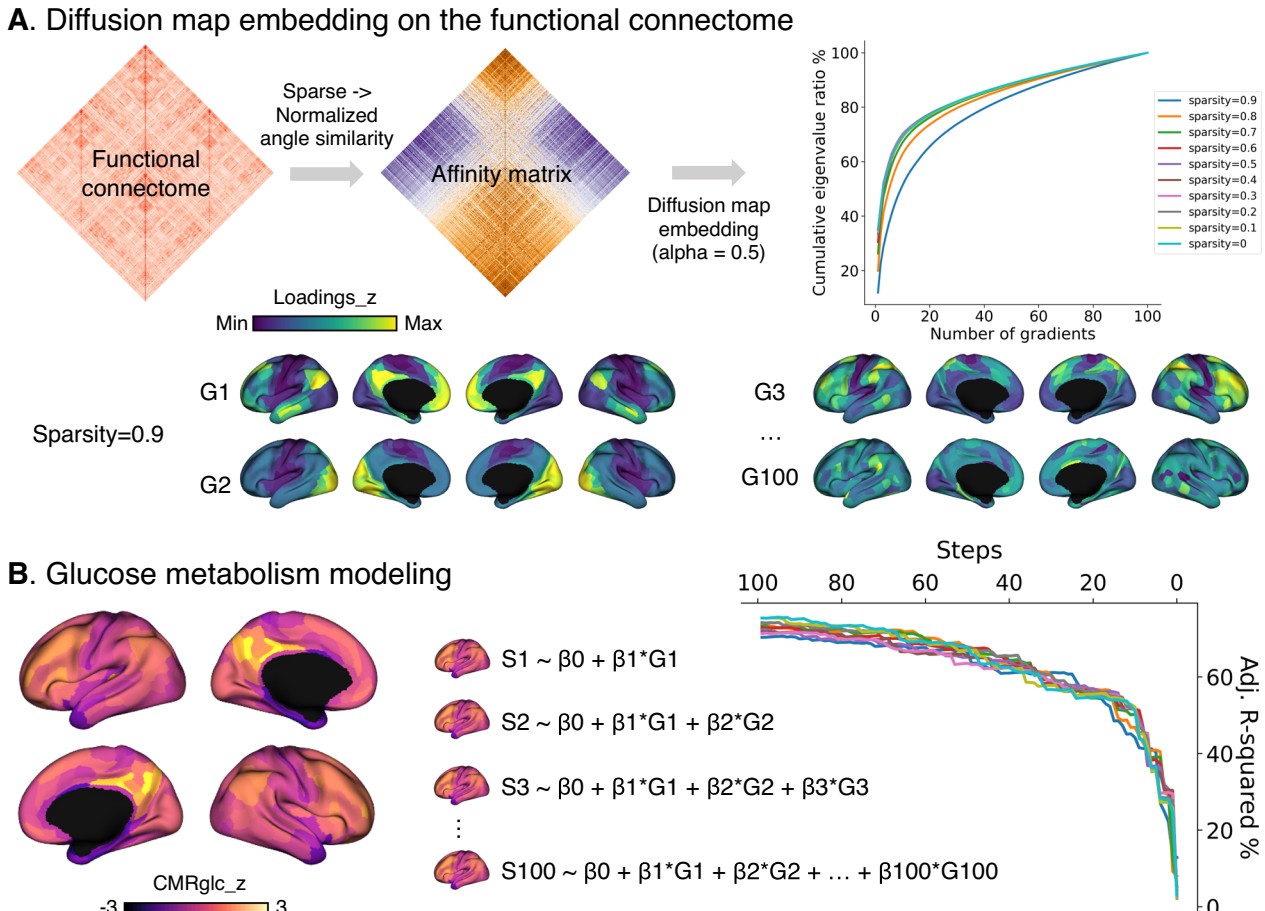

**Fig. 2 | Generating functional organization gradients and glucose metabolism maps. A** Illustrates the decomposition processing for the group-level functional connectome. Each sparsity parameter means the threshold of the functional connectome, for example, the top 10% uses sparsity = 0.9, and the full connectome uses sparsity = 0. **B** Shows the group-level CMRglc z-scored map and how the regression models are generated. The chart displays the adjusted R-squared values of the model by entering more gradients.

performed with quantitative [18F]FDG-PET at the Technical University of Munich with 9 individuals as experimental datasets (4 females; age: 43 ± 7 years) and 11 as replication dataset (6 females; age: 27 ± 5 years). We analyzed the CMRglc map and functional connectome, summarized into 360 parcels by a multimodal parcellation (MMP) scheme[43].

Group-level functional organization gradients were computed by reducing the dimensionality of the affinity matrix of the mean functional connectome across individuals using diffusion map embedding (Fig. 2A). We computed the 100 principal gradients for 10 sparsity parameters each by thresholding the top 10% (sparsity = 0.9) to the full (sparsity = 0) functional connectome. The sparser the matrix, the higher the variance explained by the gradient framework. For brain gradient maps with sparsity = 0.9, the standard setting of previous works[13,20,21,44–46], G1 separated association areas from the visual cortex, G2 separated somatomotor cortex from the visual cortex, and G3 separated task-positive from task-negative networks. The G4-20 brain maps are shown in Supplementary Fig. S1. The more similar the gradient loadings of different regions, the more similar the FC profile of those regions. For gradients 4–100, the residuals gradually decreased and included more subtle topographic variance. There was no intercorrelation among these gradients (Supplementary Fig. S2), indicating their statistical independence. The group-level CMRglc map (Fig. 2B) was calculated by averaging the individual CMRglc maps and then z-scoring both CMRglc and gradient maps. The map shows a spatial differentiation of glucose uptake, with one anchor in the precuneus and the other in the temporal pole regions.

We then generated the linear regression models without removing variables, as shown in Fig. 2B. Step 1: CMRglc = β0 + β1*G1, step 2:

CMRglc = β0 + β1*G1 + β2*G2, and so on until step 100: CMRglc = β0 + β1*G1 + … + β100*G100. We plotted how the adjusted R-squared values would increase or decrease as more gradients entered the model. It showed that the adjusted R-squared increased logarithmically with the number of gradients. For the last step (step 100), the regression model explained over 75% of the variance of the CMRglc map (sparsity ranging from 0 to 0.9: 76.2, 76.3, 74.4, 75.8, 75.8, 77.1, 75.5, 76.3, 77.0, and 77.7%). Moreover, the regression model could explain over 40% of the variance of CMRglc with only the first 10 gradients (sparsity from 0 to 0.9: 45.4, 46.7, 47.7, 48.7, 48.7, 48.9, 49.4, 49.1, 47.8, and 44.3%). Together, these results suggest that the way the brain's intrinsic functional networks are spatially organized is directly related to how much energy a given region uses. That is, regions with similar connectivity profiles consume similar amounts of energy, extending previous notions of connectivity and energy expenditure to a spatial framework of brain organization.

### Model explanation and regularization
Next, to determine how dense (i.e., sparsity) the FC can show a best CMRglc-gradients reconstruction, we plotted the CMRglc explanation by gradients (y-axis) and accumulated variance ratio of FC gradients (x-axis) in Fig. 3A. Even though different sparsities of FC can reach >70% finally CMRglc reconstruction, sparsity = 0.9 reached faster than other numbers. For example, we found under the 50% of cumulative eigenvalue ratios, the gradients from sparsity = 0.9 can reach around 35.7% variance of CMRglc, and other sparsities were around 20%. The performance of sparsity = 0.9 was above the other numbers. This difference suggests that weak connectivity may be less useful to reconstruct metabolic structure. Given that

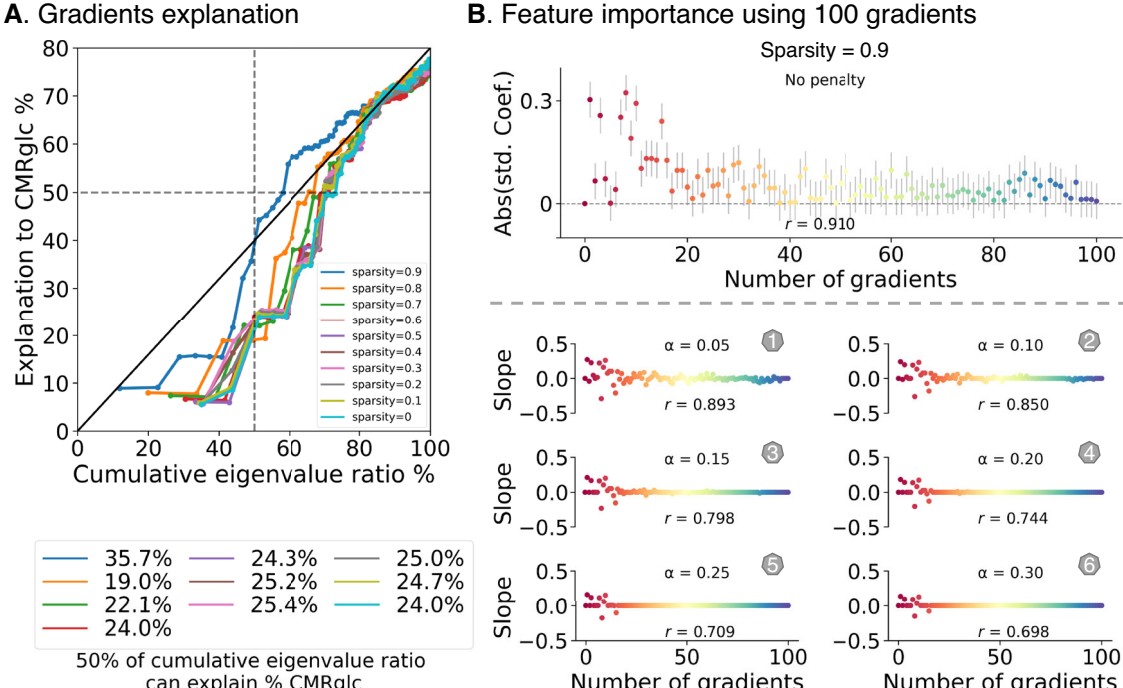

**A**. Gradients explanation

**B**. Feature importance using 100 gradients

**Fig. 3 | Model explanation and regularization. A** The x-axis showing cumulative eigenvalue ratios (each gradient eigenvalue divided by total eigenvalue) and the y-axis showing CMRglc variance explained by gradients. The lower panel further displays how their relationship (slope) changes by model with more gradients entered (starting from 10). **B** Plots scatter using elastic net regularization (lasso/ ridge = 1) onto the step 100 model: one without penalty (standardized regression coefficient and 95% confidence interval by number of gradients), and the other with six penalty parameters ($\alpha$) from 0.05 to 0.3 (slope by number of gradients). Pearson correlation coefficients ($r$) were calculated using true CMRglc and model-predicted CMRglc maps.

---

using sparsity = 0.9 returned the best symmetry and explanation of both the functional connectome and CMRglc map, the following analyses were performed for sparsity = 0.9.

Next, we used a statistical method called elastic net to simplify our model and test how the different brain gradients are related to energy use in the brain. By adjusting the model's complexity, we could investigate how much each gradient contributed to predicting brain energy use. We set the L1 ratio to 0.5 to balance the lasso and ridge regularization and manually adjusted penalty parameters ($\alpha$) from 0.05 to 0.3 with 0.05 intervals for step 100 (Fig. 3B). We first plotted the absolute values of standardized regression coefficients with a 95% confidence interval (CI) to reveal the relative contribution of each gradient to the fit without including a penalty. The signed values can be seen in Supplementary Fig. S3. The top five regression coefficients were: G8 [−0.323(−0.374, −0.273)], G1 [0.303 (0.253, 0.354)], G10 [0.242 (0.248, 0.343)], G3 [0.257 (0.205, 0.307)], and G7 [0.252 (0.201, 0.302)] with the association between true CMRglc and model predicted CMRglc to be 0.910 (Pearson $r$). With increasing penalty parameters, the Pearson $r$ between true CMRglc and model-predicted CMRglc was reduced, and the first gradients were still contributable, indicating that the order of gradient may follow the energy consumption principle, i.e., accumulative gradients by the order. This illustrates that low dimensions, explaining a lot of variance in intrinsic functional dimensions, also a lot of regional variance in energy maps, further underscoring the link between functional topology and brain metabolism.

**Validation using null models**
We further tested whether the gradient maps (sparsity = 0.9) could predict a surrogate CMRglc map of random distribution generated by variogram (Fig. 4A). Considering spatial autocorrelation[36,47–49] of brain maps, we used a variogram to permute the brain maps based on the geometric distance matrix of central coordinates of the Glasser atlas. Therefore, it can generate spatially dependent surrogate maps on the target map. We permuted the CMRglc map 1000 times and obtained 1000 CMRglc surrogate maps. We

observed that the true model exceeded 99.4% of the variogram auto-correlated models. We also permuted the gradient maps 1000 times using the variogram method, which showed that the true model exceeded all the variogram null models (Fig. 4B). This indicates that the gradient-energy relationship is generated by the "true" CMRglc map and not by random spatial patterns, providing further validity for our observations.

**Asymmetry modeling**
Next, we investigated whether the relationship between functional organization and glucose metabolism is consistent across hemispheres or shows hemispheric specificity. We calculated functional gradients in the left and right hemispheres separately, with right hemisphere (RH) gradients aligned to left hemisphere (LH) gradients using Procrustes rotation (Fig. 5A). We then calculated asymmetry maps (LH-RH) for both functional gradients and CMRglc, with CMRglc values z-scored within each hemisphere (Fig. 5B).

We trained models on the LH data and then used the trained parameters to predict CMRglc in the RH and vice versa for another hemisphere (Fig. 5C). We then defined five model variants to compare hemisphere-specific versus cross-hemisphere prediction (Fig. 5D, E): mode 1 (LH-RH specific): train and predict within own hemisphere; mode 2 (RH-LH specific): train and predict with exchange of hemisphere; modes 3–4 (generalization models): train on one hemisphere predict the both; mode 5 (asymmetry-based): train on gradient asymmetry to predict CMRglc asymmetry. We evaluated predictions using covariance and slope metrics (Fig. 5F).

Hemisphere-specific models showed modest performance (mode 1: covariance = 0.058, slope = 0.603; mode 2: covariance = 0.041, slope = 0.424), while cross-hemisphere models performed substantially worse (modes 3–4: covariance < 0.011, slopes < 0.12), with slopes indicating near-constant predictions. The asymmetry-based model showed intermediate performance (covariance = 0.051, slope = 0.530). Predicted maps are shown in Supplementary Fig. S4. We could also replicate the finding using joint alignment between left and right hemispheres (Supplementary Fig. S5),

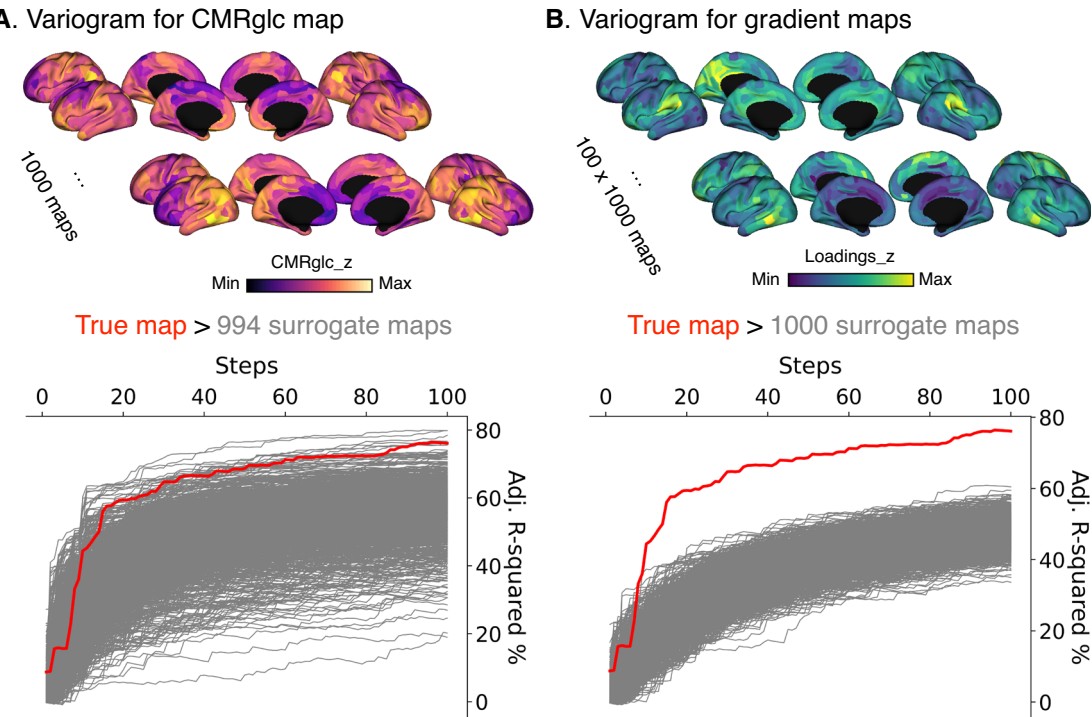

**Fig. 4 | Null models autocorrelated with surrogate maps (sparsity = 0.9). A, B** Show 1000 surrogate maps for the CMRglc and gradient maps using variogram spatial autocorrelation based on the geometric distance matrix. Red and gray lines indicate true and null models.

which was qualitatively similar to Procrustes alignment. In particular, we found the variance for the modes 1 and 2 was 0.059 and 0.082, but for the modes 3, 4 and 5 was 0.019, 0.043, and 0.053. The slopes for the modes 1–5 were: 0.616, 0.856, 0.208, 0.448, and 0.550.

Overall, these results suggest some degree of hemispheric specificity in energy-function relationships. However, several limitations warrant caution. First, even the best models explain only half of metabolic asymmetry (slope = 0.530). Second, the less fitting of cross-hemisphere generalization may partly reflect methodological challenges in spatial alignment across hemispheres. Third, our z-scoring procedure normalizes away absolute hemispheric differences, focusing on relative spatial patterns within each hemisphere. We therefore interpret these findings as preliminary evidence for hemispheric specificity, acknowledging that weak predictive performance and potential methodological confounds limit strong biological conclusions.

### Individual level analyses

To assess whether our observations of the entire cortex and asymmetry extend beyond the group level, we tested the models at the individual level (N = 20). We used the raw individual gradients for the whole cortex modeling (Supplementary Figs. S6–11). Overall, our group-level findings were also present at the individual level. Specifically, for the reconstruction models, the adjusted *R*-squared values increased as more gradients entered the CMRglc map for all subjects, and the mean explained variance was 44.6% with a standard deviation of 7.2% (Supplementary Fig. S6). Four individuals' gradient maps (Supplementary Fig. S7) and CMRglc (Supplementary Fig. S8) maps were shown to illustrate the robustness of our model at the level of the individual. The accumulated explained variance of gradient decomposition had comparable levels for all subjects (mean MAE = 0.017, standard deviation = 0.017, Supplementary Fig. S9). Regarding the individual cross-hemispheric validation, the mean performance levels of RH-to-LH and LH-to-RH were highly similar, but they showed high variability at the individual level (Supplementary Fig. S10). Regarding the comparison between competition and lateralization modes, competition performed better than lateralization for 80% of the individuals (Supplementary Fig. S11).

### Discussion

In this study, we investigated how the functional organization of the brain is related to its energy consumption. We focused specifically on regional energy consumption using PET imaging of glucose metabolism. It was hypothesized that the systematic functional organization of the brain is directly related to regional variability in energy consumption. To test this, we used a linear regression model in which brain energy is predicted by functional gradients representing unique global patterns of brain intrinsic co-activation. Our model explained approximately 70% of the variance in energy consumption at the group level, a robust result that held across different levels of sparsity in the FC matrix. Interestingly, gradients generated from sparser FC networks were better at capturing regional variability in energy expenditure, suggesting that the strongest functional connections in each region play a key role in determining how much energy the brain consumes. We also used these models to investigate how differences in energy use between the two hemispheres of the brain relate to their functional organization asymmetry. By comparing hemispheric-generalization and specific models, we found that energy asymmetry was better explained when the hemispheres were treated as separate systems, yet the overall comparative effect was weak. Our findings held when tested at the individual level, further confirming the robustness of our results. Overall, our study reveals an intrinsic link between the spatial organization of intrinsic brain function and its energy expenditure, providing new insights into the energetic basis of brain function and how the brain manages its resources.

Our first step was to examine whether regional variability in energy expenditure can be explained by intrinsic functional organization gradients. By accumulating organizational gradients in our model, the first 5 gradients already outperformed the predictive power of models based on graph topological metrics (Pearson *r* ranging from 0.41 to 0.51). Including 60 gradients captured more than 70% of the variance. Previous studies have shown moderate correlations between energy intake distribution and brain function, with Pearson *r* values ranging from 0.3 to 0.5[8,10–12,50]. Our model, which uses the topography of intrinsic brain function, achieved a stronger relationship. As the gradient number increased (e.g., higher-order latent dimensions), their spatial composition became more local. Notably, gradients based on the top 10% of the connectivity synchronized strongly with

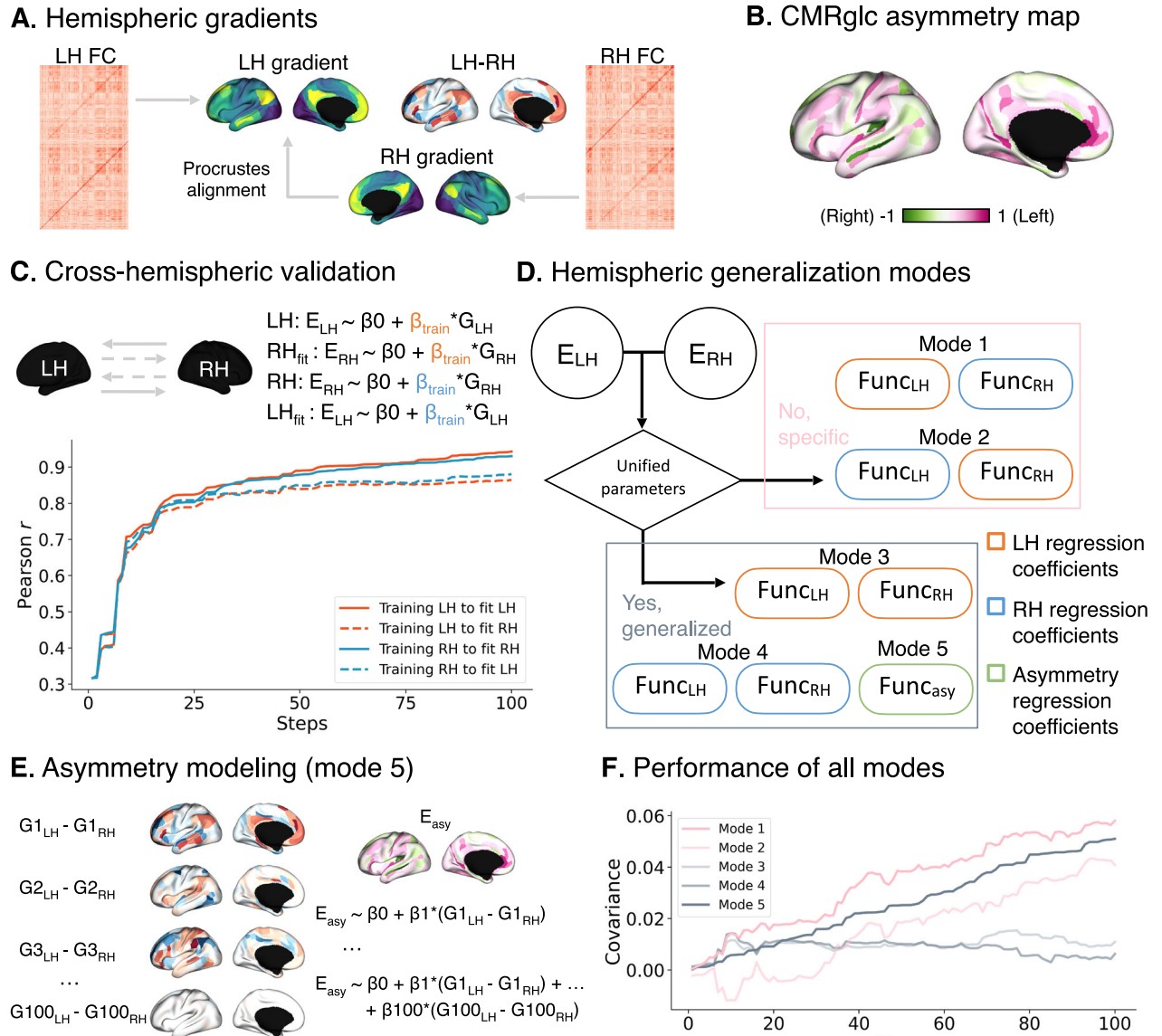

**Fig. 5 | Asymmetry modeling. A** Illustrates how hemispheric gradients are calculated and an example asymmetry map of G1. **B** Shows the CMRglc asymmetry map. **C** Visualizes the cross-hemispheric validation for the models and performance in the training and fitting sets. **D** Illustrates five optimization modes depending on whether LH and RH share the energy-gradient coefficients. If not, modes 1 and 2 use independent LH and RH training coefficients themselves (mode 1) or exchange (mode 2). If yes, modes 3, 4, and 5 share the LH (mode 3) or RH (mode 4) or asymmetry (mode 5) coefficients. **E** Visualizes how to calculate hemispheric differences after the modes above, e.g., the lateralization modeling (mode 5). **F** Compares fitting covariance between specific (modes 1 and 2) and generalization (modes 3, 4, and 5) models. All mode predicted brain maps can be seen in Supplementary Fig. S4.

the energy map in terms of variance explained in the FC matrix and the energy consumption map. Using more information from the functional connectome (i.e., less sparse) did not improve the model fit to the energy map. This suggests that regional energy consumption strategies may most closely link to the strongest connections in the functional connectome and less so to weaker connectivity patterns.

Although we had excellent group-level results of model fit, there was still large individual variation (the adjusted $R^2$: 32–58%). The reasons might be explained by individual differences in glucose usage types and lack of geometric distance constraints. However, direct biological evidence would be needed to establish that perspective. Geometric distance might constrain the FC[40]. Gaining a functional distance[51] might then improve individual prediction. However, individual data quality and heterogeneity might also play a role.

Conceptually, weak connections may not consume as much energy (reduced impact on glucose metabolism). One possible explanation could be

that hub regions consume more energy than non-hub regions[10] and that energy consumption of weak connections is influenced by stronger connections, such as hub regions[52], in an "energy hierarchy". Indeed, functional hubs serve as connectors between different brain modules or within each module[52,53]. For example, the precuneus has been identified as a hub region[52,53] and is also consistently a high energy expenditure region in our study, also seen in previous research[9,11,50]. Another factor that may explain the observed patterns may be redundant connections. A non-redundant model of energy expenditure suggests that sensory areas show redundant connectivity, while parietal and frontal areas show non-redundant connectivity[54]. This is consistent with the energy expenditure map observed in our study, as parietal and frontal cortices also have high energy expenditure, and sensory areas have low energy expenditure. By aggregating the connectivity per region, previous work found a correlation with the energy expenditure map (correlation coefficient around 0.5)[8]. Identifying the principle or threshold of redundant connectivity, in terms of energy expenditure, could improve our

understanding and interpretation of the energy expenditure map. For example, different thresholds could be applied to different brain regions. Overall, an energy-driven understanding of brain function may provide novel perspectives on the relationship between computational demand and energy supply in the human brain across the lifespan.

Brain lateralization has been proposed as a mechanism for optimizing brain energy requirements[22]. Here, we evaluated this hypothesis by using asymmetry maps of functional organization to model the energy asymmetry map. We found that hemispheric specific models outperformed the generalization models with a small winning rate of 14%. This suggests that energy asymmetry may not result solely from cumulative functional lateralization (generalization models) in topology. While asymmetry of functional gradients has been extensively studied[15,17,18,20,21], our results suggest that gradients may not only reflect static brain organization but also how the brain operates across different spatial manifolds in the form of independent hemispheric optimization in the context of competing for energy source. Such an interpretation could be in line with conceptualizations that functional lateralization results from interactions between hemispheric functions[55,56]. For example, hemispheric competition describing the changing role of hemispheres has been observed during attention and sleep over time[57–60]. Our findings give a novel insight that energy lateralization may be the result of such functional competition across spatial manifolds in addition to temporal scales.

There are also limitations to this study. First, although we found individual variability, the reasons are certainly more complex than biology and demographics. Future studies could enhance the sample size and add further features of interest, such as genetics (using twin models) and psychological factors (mental health). Second, FC is more stable with more time series obtained, such as the Human Connectome Project (1200 time series). Increasing the time series will further help identify the redundancy in functional organization informed by glucose metabolism. Last, regarding the comparisons between hemispheric generalization models, we simply applied parameters trained on the left or right hemisphere. More complex inter-hemispheric models might provide a more comprehensive result.

In conclusion, here we revealed how cortical energy expenditure is related to spatial maps of functional brain organization. In doing so, we provide a novel framework for functional organization in the context of its energy landscape, illustrating how the topological organization of intrinsic function may relate to the optimization of metabolic processes. These insights may hold relevance for a basic understanding of the biological basis of intrinsic functional activity in the human brain and provide further perspectives for its changes in neuropsychiatric and neurological disorders.

## Methods
### Data source
We reanalyzed the data from OpenNeuro (data number: ds004513). Briefly, data collection was conducted at the Technical University of Munich with 9 individuals as experimental datasets (4 females; age: 43 ± 7 years) and 11 as replication datasets (6 females; age: 27 ± 5 years). Imaging data were acquired on an integrated PET/MR (3T) Siemens Biograph mMR scanner (Siemens, Erlangen, Germany) and used a 12-channel phase-array head coil for the MRI acquisition.

Regarding the PET data collection, they were collected in list-mode format with an average intravenous bolus injection of 184 MBq (SD = 12 MBq) of [18F]FDG. In parallel to the PET measurement, automatic arterial blood samples were taken from the radial artery every second to measure blood radioactivity using a Twilite blood sampler (Swisstrace, Zurich, Switzerland). The [18F]FDG-PET includes 33 dynamic frames: $10 \times 12$ s, $8 \times 30$ s, $8 \times 60$ s, $2 \times 180$ s, and $5 \times 300$ s. The attenuation correction was based on the T1-derived pseudo-CT images. The [18F]FDG-PET images were motion-corrected, spatially smoothed (FWHM = 6 mm), and partial volume corrected using the gray matter (GM), white matter (WM), and cerebrospinal fluid (CSF) masks derived from the T1 images. Arterial blood samples were converted to plasma and modeled as a sum of three exponential functions. The arterial input function was calculated by evaluating

this function at the times from the PET dynamic frames. The net uptake rate constant (Ki) was estimated using Patlak's plot based on the last five [18F]FDG-PET preprocessed images and the arterial input function. The CMRglc map was calculated by multiplying Ki by the plasma glucose concentration and then dividing by the [18F]FDG lumped constant. Finally, the individual CMRglc map was registered to MNI152NLin6ASym 3-mm template and volume to surface mapped to mid-thickness of fsLR-32k space and summarized into 360 parcels using MMP[43].

The fMRI data were acquired during a 10-min time interval using a single-shot echo planar imaging sequence (300 volumes; 35 slices; repetition time, TR = 2000 ms; echo time, TE = 30 ms; flip angle, FA = 90°; field of view, FOV = $192 \times 192$ mm²; matrix size = $64 \times 64$; voxel size = $3 \times 3 \times 3.6$ mm³). Anatomical images were based on a T1-weighted 3D-MPRAGE sequence (256 slices; TR = 2300 ms; TE = 2.98 ms; FA = 9°; FOV = $256 \times 240$ mm²; matrix size = $256 \times 240$; voxel size = $1 \times 1 \times 1$ mm³). Functional images were slice-time–corrected, realigned, motion-corrected, skull-stripped, and registered to the anatomical images. Thereafter, the global mean intensity was normalized across the fMRI run, the nuisance signals were regressed out (scanner drift, physiological noise, and head motion signals), and the time series were band-pass-filtered (0.01–0.1 Hz). The regression of the nuisance signals modeled the scanner drift using quadratic and linear detrending, whereas the physiological noise was modeled using the five principal components with the highest variance from the decomposition of white matter and CSF voxel time series. Motion regressors were derived from rigid-body realignment during preprocessing, including three translational (X, Y, Z) and three rotational (pitch, yaw, roll) movements. Then, the functional image was spatially smoothed (Gaussian filter, FWHM = 6 mm) and registered to the MNI152NLin6ASym 3-mm template through the anatomical image. Finally, the volume to the surface function was used to map the time-series volumetric data to the mid-thickness of the fsLR-32k space and summarized into 360 parcels using MMP. The functional connectome of each individual was calculated by Fisher-z transforming the time series correlation matrix.

### Gradients of functional organization
After obtaining the individual functional connectome, we calculated the affinity matrix in a normalized angle with different sparsity parameters from 0 (full connectome) to 0.9 (top 10% connectivity of the connectome). Normalized angle correlation refers to a technique where the correlation between two vectors is calculated by first normalizing the vectors (making them unit vectors) and then measuring the cosine of the angle between them. Next, we employed nonlinear dimensionality reduction to generate 100 principal gradients for each individual and at the group level. At the group level, we set the left hemisphere (LHLH) connectivity gradients as the reference template and aligned the right hemisphere (RHRH) connectivity gradients to this template using Procrustes rotations. This procedure optimally rotates, translates, and scales the right hemisphere gradient space to minimize differences with the left hemisphere template, ensuring that corresponding gradients across hemispheres represent comparable organizational patterns rather than arbitrary mathematical dimensions. In addition, to provide a multi-view of the alignment, we used joint alignment between LH and RH for replication. This approach computes LH and RH gradients simultaneously in a shared embedding space using concatenated LH and RH connectivity matrices[36,61]. Gradients reflect the eigenvectors, and gradient loadings reflect the eigenvalues. Regions with minimal connectivity are similarly farther apart (segregation). Regions having similar connectivity profiles are embedded together along the eigenvector (integration). All steps were accomplished in the Python package Brainspace[36]. The name of this diffusion-embedded mapping, which belongs to the family of graph Laplacians, is derived from the equivalence of the Euclidean distance between points[13,36,62]. It is controlled by a single parameter $\alpha$, which controls the influence of the density of sampling points on the manifold ($\alpha = 0$, maximal influence; $\alpha = 1$, no influence). Based on the previous work[13], we followed recommendations and set $\alpha = 0.5$, a choice that retains the global relations between data points in the embedded space and has been suggested to be relatively robust to noise in the affinity matrix.

## Asymmetry

To quantify the inter-hemispheric differences, we calculated asymmetry by LH minus RH. MMP offers an atlas of homologous regions between hemispheres, allowing asymmetry to be calculated. Regarding the CMRglc, we z-scored LH and RH separately and then calculated the energy uptake asymmetry map. Regarding gradients, we computed the LH and RH gradients, then aligned the RH to the LH gradients to make them comparable in the main figures. In addition, we also aligned LH to RH gradients to test the alignment robustness, shown in Supplementary Figs. Finally, we calculated the gradient asymmetry maps.

## Statistics and reproducibility

We then generated the reconstruction models to observe how the explanation of variance (adjusted $R$-squared) changed along the steps at the group level ($n = 20$). We set the CMRglc map as a dependent variable and every gradient as an independent variable. Ordinary least squares estimation in the Python package statsmodels (https://www.statsmodels.org/stable/index.html) was used to generate the models. We extracted 100 gradients, resulting in 100 steps or regression models. Adjusted $R$-squared is a modified version of $R$-squared that has been adjusted for the number of independent variables in the model. Adjusted $R$-squared $= 1-[SS_{residual}/(n-k)]/[SS_{total}/(n-1)]$, where $n$ indicates brain data point (i.e., 360 for MMP) and $k$ indicates the number of gradients in the model. Adjusted $R$-squared penalizes model complexity and can decrease when adding variables that contribute little predictive power, even when predictors are orthogonal to each other. While gradients are orthogonal by construction in the embedding space, a decrease in adjusted $R$-squared when adding higher-order gradients indicates that these gradients have weak relationships with glucose metabolism and do not improve prediction.

In addition, we regularized the step 100 model using balanced Lasso and ridge algorithms. We manually tuned the penalty parameter ($\alpha$) from 0.05 to 0.3 to visualize which gradients can be regularized and how much it can influence the model fit. The range of $\alpha$ values was chosen because no gradients could be selected with $\alpha$ value greater than 0.3 and no gradients could be unselected with $\alpha$ value lower than 0.05.

The reconstruction was done using the group-level gradients and the CMRglc map and further generalized for each individual separately for the replication ($n = 20$). The group-level gradients were calculated from diffusion map embedding on the group mean FC matrix. This group-level and individual-level analysis framework was also employed in asymmetry findings.

## Reporting summary

Further information on research design is available in the Nature Portfolio Reporting Summary linked to this article.

## Data availability

All raw and preprocessed resting state fMRI and PET data are available through OpenNeuro[8,63], which can be downloaded at https://openneuro.org/datasets/ds004513/versions/1.0.4. In addition, the group-level and individual-level FC matrix and CMRglc maps are available at the above GitHub repository, which ensures reproducible results with the scripts.

## Code availability

All analyses were conducted based on Python 3.8, and the main dependencies are BrainSpace (https://brainspace.readthedocs.io/en/latest/index.html) and statsmodels (https://www.statsmodels.org/stable/index.html). Scripts for this study are shared at a GitHub repository (https://github.com/wanb-psych/asymmetry_energy).

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

## Acknowledgements

The concept of this study was inspired by the geometric constraints on brain function paper[40]. B.W. is supported by the International Max Planck Research School on Neuroscience of Communication: function, structure, and plasticity (IMPRS NeuroCom). V.R. is supported by the European Research Council (ERC) under the European Union's Horizon 2020 research and innovation program (ERC Starting Grant, ID 759659). S.L.V. is funded by the Lise Meitner and Otto Hahn Award at the Max Planck Society and Helmholtz International BigBrain Analytics and Learning Laboratory (HIBALL), supported by the Helmholtz Association's Initiative and Networking Fund and the Healthy Brains, Healthy Lives initiative at McGill University.

## Author contributions

Bin Wan: conceptualization, methodology, formal analysis, writing-original draft, writing-review and editing, visualization, project administration,

funding acquisition. Valentin Riedl: data collection, writing-review and editing. Gabriel Castrillon: data collection, writing-review and editing. Matthias Kirschner: writing-review and editing. Sofie L. Valk: writing-original draft, writing-review and editing, supervision, and funding acquisition.

## Funding

## Competing interests

The authors declare no competing interests.

## Additional information

**Supplementary information** The online version contains Supplementary material available at https://doi.org/10.1038/s42003-026-09693-w.

