## [Transparent Peer Review File · Communications Biology]

Bridging Glucose Metabolism and Intrinsic Functional Organization of the Human Cortex

Corresponding Author: Mr Bin Wan

Version 0:

Reviewer comments:

Reviewer #1

(Remarks to the Author)

The central idea of the paper is to use connectivity gradients as independent variables to account for the spatial pattern of energy consumption maps obtained using PET. This is done both on the whole brain level and by contrasting left and right hemispheres.

Please note that this review was conducted with the assistance of a trainee reviewer, whose comments are also provided.

The invited reviewer comments:

As the length of my recommendations list may suggest, I do not think the paper is strong methodologically. On too many instances the authors report trivial findings and reveal some lack of insight on the methods they use (unless I misunderstood what has been done – the lack of details in Methods section does not help this). The emphasis on looking at the broad range of gradients, in my view, is also problematic methodologically, see details below. It would also be good to get more clarity on whether this association is very specific to gradients. E.g., if, instead of gradients, I would use ICA-derived functional networks (DMN etc), would I fail to account for metabolism in a similar way?

Introduction

The first paragraph – I felt as if the authors speak their own language here. I think I understood what they meant (apparently, contrasting local functional characteristics vs long-distance connectivity), but the message here is not transparent if not convoluted. E.g., the measures such as region's centrality is a characteristic derived based on connectivity (ie., it can also be viewed as "an outcome of inter-regional connectivity" or "topological organization"), hence, the distinction between the two views is not clear. The meaning of "systematically organized by network community integration and segregation" is also non-transparent.

"Functional gradients are reproducible.." - of relevance for the further analyses : only the first two/three or generally speaking?

"We hypothesize that there is a relationship between topological organization of intrinsic functional connections and energy expenditure, as 1) the organization of the cortex is likely a result of evolutionary pressures on minimizing energy demands 8,30,31 and 2) regions that have a similar functional organization likely have a similar energy expenditure 13. " –not sure that this is well motivated, I don't think Ref 13 is a ref that demonstrates this.

"not how much energy functional organization consumes" – not sure what functional organisation's energy consumption means.

Figure 1 – I found it very confusing, seems to be a mixture of preprocessing and reasoning blocks. 1) in upper right plane – looks very *grounded* but what does it have in common with the analyses? Are you going to reconstruct all these ATP, ADP, O2 etc? I recon an FDG-PET image would be more fitting here. 2) Equation is likely to be lacking a term for the intercept. The equation seems to be equivalent to "Reconstruct?", why they are not connected then? 3) I don't understand what "Regions A and B share etc" box does in between "similar" and "decompose"

“The first aim of our study.. “ – I expected to find a second aim somewhere further in the text, but this was not the case.

“We hypothesize that when using gradient maps” – it’s the third time throughout the Intro when the authors hypothesise something, perhaps the authors need to review this.

“the model performance should increase step by step with an increasingly detailed description of functional brain organization (e.g. increased number of gradients and variance explained).” – it will be true even if ones starts adding “random” predictors, I guess the authors mean that it will increase in some statistically meaningful way.

“under the assumption that the spatial pattern of asymmetry” – is this an assumption or a hypothesis that you want to test using gradients as IVs?

Methods.

The method section needs to considerably expanded to include key details. With them lacking, I am not able to provide a coherent recommendation for this section. However, certain processing choices seem to me questionable or at least need a better justification/explanation. Here are some points.

“The images were motion-corrected, high-pass (0.01-0.1 Hz) filtered, regressed out GM, WM, and CSF signals” - does this represent the exact order of preprocessing stages, and if yes this is a bit unorthodox. What does motion-corrected exactly means (e.g., how the motion regressors were constructed).

Why PET images could not be registered via native surface templates as it was done to fMRI data, implying differences in (accuracy of) registration to the template space and double interpolation in the volumetric space for PET data?

If there were native surface spaces and tissues segmentations available, I presume there exist structural volumes for each subject. How the mapping between PET/fmri and structural data was handled?

Why different smoothing kernels were chosen to be different between fMRI and PET data? Why was a large (10 mm) smoothing kernel needed for functional data? I believe this will be of a direct relevance for the results, as this will induce spatial dependencies and therefore induce gradient-like features in the data.

Why parcellation had to used? Does it not directly contradict the concept of “gradient” that presumes there are no regional “borders” in the brain but only smooth transition? I guess this is the reason why we call them “gradients”. Otherwise, I would call it by what it precisely is, a non-linear embedding of regional connectivity features.

“time series of signal-to-noise ratio (SNR)” – what’s that? I always thought SNR is a mean divided by standard deviation, what is the timeseries thereof?

I would recommend presenting the key information on the imaging parameters here, not by a reference to some other paper.

“the affinity matrix in a normalized angle” –What is it, a normalised angle? Was it a name for a transformation applied to convert matrix of z-transformed correlations into affinity matrix?

“Finally, the comparative individual functional gradients were assessed.” – what does “comparative” mean in this context? More details on Procrustes alignment is required, because it is not clear what exactly has been aligned (it’s not matching spatial x,y,z coordinates, correct?)

“suggested to be relatively robust to noise in the covariance matrix.” – how relevant is this? You have a correlation matrix, obtained after heavy temporal and spatial filtering and averaging across vertices within each parcel. What noise are you referring to here?

Stepwise regression – not sure why the emphasis is on stepwise regression. Since the gradients are mutually orthogonal, the authors could just compute the full model, calculate the unique variance per each term and their cumulative sum up to a required component order.

(After reading the Results section where some addition details of analyses could be found, I have a further comment to the last point. I suspect non-orthogonal gradients can indeed emerge as a result of averaging across subjects (provided this was the case, I hope I understood this correctly). But then how were the gradients matched across subjects. Don’t you trivially confuse the gradient order with gradient identity? As an example: assume that in one subject the order of the first (unimodal-transmodal) and the second (sensorimotor-visual) canonical gradients was flipped in one subject but not the others, would you still continue with averaging across subjects without re-ordering (I hope not). As the gradient order increases, this will for certain lead to decreasing across-subject consistency, and therefore an average of a high-order gradient represents a random mixture.)

“Adjusted R-squared decreases or is negative when adding more gradients.” – this certainly is not universally true.

“(i.e., 360 for MMP and 8856 for fsLR-5k)” – I got confused. I thought all analyses were done using Glasser parcellation (or

some weren't?)

"We manually tuned the penalty parameter from 0.00005 to 0.0003 to visualize.." – what is the criteria for manual tuning. Was it done on subject-by-subject basis?

"z-scored LH and RH" – z-score calculated across subjects or a map?

"then aligned RH to LH gradients to make them comparable in the main figures" – what is the procedure for the alignment? (looks like more details were presented in Results, but it needs to be more explicit here)

Results

"There is no intercorrelation among these gradients (Supplementary Figure S2), indicating the statistical independence of gradients." – See my comment for stepwise regression above. My understanding is that gradients represent spatial weights on eigenvectors of a Laplacian matrix, i.e., they are orthogonal by construction. However, if the authors are worried that non-independence could emerge from averaging gradients across subjects, then I don't understand how the authors managed to match them across subjects, as it cannot be assumed that the gradients with the order number N in two subjects are *same*.

"to determine whether the regional variance of FC and CMRglc is symmetrically explained by the gradients" – this needs to be better motivated, why it is important to look at FC variance?

"we used slope and mean absolute error" – what does mean absolute error tell us?

"manually adjusted penalty parameters" – I think "manually" is confusing in this context. If I understood this correctly, the authors computed regularised regression for a range of penalty values, what is "manual" about it?

"With increasing penalty parameters, the Pearson r between true CMRglc and model-predicted CMRglc was reduced" – of course it will be, as the heavier penalty will reduce an effective degree-of-freedom of the fitting model. I guess it would be better if - instead of reporting this - the authors work out quantitatively the optimal penalty.

"This illustrates that low dimensions that explain a lot of variance.." the sentence contains both interesting and trivial findings. The fact that the strongest (spatially distributed) gradients account for the metabolism is interesting. The fact that weakest gradients account less for metabolism is most likely to be trivial. I presume, as the order number of a gradient increases, their spatial composition becomes more and more local, in effect, accounting for the local deviations in metabolism (the relationship which potentially can be induced by both intrinsic and applied smoothing in the data).

"variogram" – I think some description in Methods is needed of what is achieved by variograms, e.g., is it a random rotation of gradients or something else?

"there was a strong correlation ($r = 0.732$ with 100 gradients)" – between reconstructed and actual map?

I am not sure I understand accurately what has been done in the "hemispheric independence" section. Here are a few points:

"we found that the variance in energy asymmetry can be explained by the combined asymmetry of gradients" – Is this a meaningful result, unlikely to be matched by random gradients with matched spatial correlation properties?

"To address this, " – to address what? Rectifying the relation between the order and predictive power?

"energy-guided gradient asymmetry maps " – not sure I understand what it is.

"but slightly worse when predicting the other hemisphere.." – unless I misunderstood this – if this is the difference in the variance accounted for in training and testing samples (=hemispheres), then this is the most trivial result.

"LH and RH use independent training coefficients" – is this fitted regression coefficients? (please clarify)

Paragraph starting with "To further determine whether LH and RH have" - My understanding of the procedure is that the authors fitted LH and RH to corresponding LH and RH energy maps. They then compared which of 4 combinations of IVs and DVs and one asymmetry fit is the best-fitting. However, I am missing a logical link on how this is informative of "competition" vs "lateralisation". The result is clearly expected based on the results reported in the above (Figure 5C vs Figure 6A).

Trainee Reviewers comments:

Whilst I feel that the overall premise of comparing BOLD response to energy consumption in the brain through the use of rfMRI and PET imaging is an interesting one, the work within this paper doesn't quite fulfil the potential that you intended it to.

Main:

As mentioned above the language used throughout the paper is somewhat colloquial. The flow of this section is disjointed and jumps backwards and forwards. Throughout this section you mention three different hypotheses may be good group together after your main background information.

On page 4, it is noted what your first aim was, however, there are no following aims after this point. Did you intend to have further aims for this work? Or was it the intention to only have one? If the later, I would consider rewording this so that you only mention that have one aim. Additionally, it would be good to separate this out from your 'Main' section to make it clear that this is the aim of your work.

Throughout this section the references (1-6, 14-16 and 20,21) that you use are 12+ years old, there are more recent references available to support your points here.

Methods/Results

The first paragraph of your results section is a very limited methodology of how you have conducted this study. The small paragraph stating you are using a secondary data source in the results section could be used within a methods section, adding total number of data sets analysed, with median and age ranges of the participants, you can then go onto say that this was subdivided into experimental and replication datasets. It would also be worth clarifying what differences there were between these subsets are if any.

Whilst I see that you have put a large methods section at the end of the paper, it would be beneficial and better placed to have a well described methods section prior to the results section, to limit confusion of what your method was throughout the paper. As this leaves the reader unclear throughout your results section about how you have carried out your analysis of the secondary data set. More of this final methods section should be condensed into a methods section prior to the results section.

Additionally, within the methods section it would be beneficial to ascertain when the data was collected and over what time period.

Parameters are mentioned as part of the initial paper for the secondary paper, are the parameters stated for fMRI or for the PET imaging too? Currently this is unclear to what parameters are noted within this original publication.

I do feel that it is good that you have broken down your report into a section beneficial to the reader.

Throughout the hemispheric independence section your method is unclear throughout. LH and RH you mention that you have trained the algorithm separately for this, it is unclear throughout this section as to what your method and thought process was.

Discussion

First sentence is clear and you should think about how this is structured and possibly stating a similar statement earlier within the work for clarity or what you are trying to convey.

You mention in paragraph one that you are investigating the differences between the hemispheres and if this is related to their energy consumption. Shouldn't this have been mentioned earlier when mentioning your aims?

Stepwise Linear Regression Model, do you have a reference for this?

Conclusion

Would benefit being in a section on its own to define this, starting with 'In conclusion...' or 'In Summary we found....'

Part of your aim in section 'Main' is to 'evaluate to what extent functional organization and energy expenditure have shared organizational features', you have loosely linked this back to the aim but could be more succinct.

Additionally, you mention that this could provide further perspectives for its change in neuropsychiatric and neurological disorders. By perspectives do you think this could instruct change in patient pathways and inform treatment plans?

Figures

There have been a lot of figures provided to illustrate your work, with supplementary material providing an additional 22 to the figures within the main text. May be beneficial to evaluate what each of the figures are adding to your work and if all are necessary to illustrate your points and what they are adding to the overall message of the work.

Reviewer #2

(Remarks to the Author)

Wan et al. present a fundamental study exploring the relationship between brain energy expenditure and the topological organization of functional connectivity. Their findings demonstrate that the spatial gradients of functional connectivity provide explanatory power in understanding the optimization of energy use in the human brain. The provided evidence supports the

study's conclusions well and contributes a novel framework for interpreting brain functional organization and asymmetry within the context of its energy landscape. Importantly, the results challenge the classical views of hemispheric asymmetry as an energy-efficient lateralized specialization. Instead, it suggests that each hemisphere operates independently, optimizing its own functional-metabolic balance rather than sharing energy resources. Below, I list my comments by reviewing the paper's different sections in order.

1 - Abstract: No comments

2 - Main (or Introduction):

While the main ideas of the study are all present in the introduction, some refactoring is needed. The hypotheses paragraph (3rd paragraph) is too long and lacks a clear, linear flow. The main hypothesis is well-stated, but the transitions between concepts—FDG-PET limitations, gradient-based modeling, and the rationale for testing 100 gradients—are abrupt. The argument would be clearer if broken into distinct sections: (1) Hypothesis and Background, (2) FDG-PET Limitations and Justification for Gradients, and (3) Rationale for Expanding Gradient Use. The sudden shift to testing 100 gradients should be introduced more smoothly, explaining why this is necessary. Improved transitions and a more structured approach would enhance clarity and readability.

The last paragraph related to brain asymmetry should be merged with the previous hypotheses paragraph, and a final concluding paragraph that highlights the main findings should be created to finalize the introduction. Overall, a refactoring of the Main section is needed.

3 - Results:

The results section is detailed and appears to stem from a robust methodology. However, the rationale behind the "Model Explanation and Regularization" section and its underlying methods is unclear. This section "Hemispheric independence" presents a well-structured comparison of hemispheric asymmetry models, demonstrating that hemispheric competition outperforms lateralization in explaining metabolic asymmetry. The clear comparisons between different energy-sharing modes strengthen the conclusion that LH and RH operate more independently than interdependently. However, one notable result—the negative covariance in Mode 2 (cross-hemisphere parameter application)—should be more clearly presented, as it indicates an inverse relationship rather than a simple failure of generalization. The quantitative comparisons across models are well-executed, but providing additional summary statistics (e.g., variance explained, R^2) alongside covariance scores would help better contextualize model performance. These adjustments would enhance the clarity of the results before interpretation in the Discussion section.

4- Discussion The discussion provides a strong synthesis of the study's key findings, particularly the robust relationship between functional gradients and energy consumption (~70% variance explained) and the importance of sparsity in functional connectivity for predicting metabolic demand. The results convincingly demonstrate that hemispheric competition better explains metabolic asymmetry than lateralization, challenging the assumption that functional asymmetry alone optimizes brain energy consumption. However, several important findings are not adequately addressed.

First, while the study includes individual-level analyses to complement group-level findings, the results show greater variability and weaker predictive power, particularly for asymmetry modeling (max adjusted $R^2 = 3.3\%$). The discussion does not acknowledge this limitation or explore why functional-metabolic coupling is less stable across individuals. Addressing whether this is due to inter-individual variability, methodological constraints, or the inherent difficulty of modeling metabolic asymmetry at the individual level would improve the interpretation.

Second, the unexpected negative covariance in cross-hemispheric predictions (Mode 2) suggests that applying one hemisphere's functional-metabolic mapping to the other results in systematic "misprediction," yet this is not discussed. It remains unclear whether this indicates true hemispheric metabolic competition—where energy use in one hemisphere negatively correlates with the other—or if it arises from modeling constraints or alignment issues. Clarifying this result would strengthen the conclusions on hemispheric independence.

Third, while the study shows that higher-order gradients do not improve asymmetry modeling, the discussion does not elaborate on why functional asymmetry does not translate into metabolic asymmetry. The results suggest that stronger functional connectivity patterns contribute more to regional energy demand, yet this relationship appears to break down in the context of asymmetry. A brief reflection on why functional gradients fail to explain metabolic asymmetry—and whether this is due to differences in hemispheric specialization, vascular factors, or the limitations of functional gradients alone—would provide a more complete interpretation. Addressing these points would enhance the clarity and depth of the discussion, ensuring that all key findings are properly contextualized.

Finally, the section in this current form does not discuss the limitations and future steps of the study.

5- Methods:

More details on the elastic net regularization approach should be provided, including the specific implementation, choice of hyperparameters (e.g., α values), and how they were optimized or selected. Further clarification on the parcellation scheme is needed, particularly regarding its resolution, whether it was symmetrically applied across hemispheres, and how hemispheric differences (if any) were accounted for in the analyses. Providing these details would improve the transparency and reproducibility of the methodology.

6 - Supplementary Material: No comments

7 - General comments:

Refining the language throughout the manuscript would improve clarity and readability. The use of non-scientific terminology

should be avoided. For example, in the Methods section, the repeated use of the term "kick out" should be replaced with a more precise term such as "removed," "excluded," or "penalized" depending on the intended meaning.

Version 1:

Reviewer comments:

Reviewer #1

(Remarks to the Author)

Whereas manuscript has improved, particularly, in explanation of methods and results, I have still considerable concerns about the unsound or redundant application of statistics. Among this: 1) Unsound analytical procedure which the authors called "model explanations and regularisation"; 2) The of generalisation of model parameters to the training set in hemispheric analysis is blown out of proportion to support statements about "hemispheric competition/lateralisation", "shared optimisation". 3) confusingly redundant procedure which the authors call the stepwise regression.

Here is a more detailed criticism:

Results

"There was no intercorrelation among these gradients (Supplementary Fig. S2), indicating the statistical independence of gradients." – Should be a trivial observation

"whether x amount of variance explained in FC by the first y gradients corresponds to x variance explained in the CMRglc map" - I think this needs more explanation, particularly, what is meant by explaining variance in FC in the current context (I hope this is not derived from eigenvalues of diffusion maps, as they do not relate to variance). Furthermore, given that gradients are computed from FC matrix and independently from CMRglc, I am not sure how valid this analysis is. With this in mind, I don't think I understand what sort of analyses are reported in this paragraph and Fig 3A.

"70% of the variance in CMRglc but only 14% in FC, a fivefold difference" - pertaining to the above, this is numerically suspicious. The gradient matrix, presumably, has to be 360 parcels by 100 gradients. (Or not?). Which means the rank of the matrix is > 25% of the number of parcels. How accounted-for variance can be lower than that is not transparent to me.

"Validation using null models" this seems to be a nice result, though – as mentioned above - the procedure that led to selection of sparsity parameter was not clear to me and hence

As a side note – isn't application of sparsity and $\alpha=.5$ for diffusion kernel amounts to double (implicit and explicit) attenuation of non-linearity of embedding? (Which is of relevance, for instance, for the accuracy of Procrustes procedure, see below)?

"Finally, we .." – "We also" would be more fitting.

"The CMRglc asymmetry map was calculated by z-scoring the LH and RH separately" – unclear motivation for separate z-scoring

As for the previous version of the manuscript, I do not understand the logic and find it extremely convoluted. I believe Fig 5C shows that FC gradients in one hemisphere accounts similarly for CMRglc in both hemispheres, with FC gradients in RH making a somewhat poorer predictions for higher number of gradients. How "energy optimization", "energy sharing", "shared optimization", "lateralisation" and "competition" follow from this is beyond my comprehension. E.g., "hemispheric competition" typically refers to a dynamic process, how this can be reflected in the association between two static features? There is no "sharing" of energy here, only generalisable modelling parameters. Who shares? Gradients? What is the logical link between "hemispheric lateralisation" (typically meaning that something is present in one hemisphere but less so in the other) and the fact that the one hemisphere gradients account for energy in both hemispheres? What "optimisation" can even possibly mean here? I feel that all this terminology is just a cover-up for a very simple, perhaps an uninspiring, result that the interhemispheric difference in energy maps are not great enough to affect generalisation of coefficients across hemispheres (which might be an inadvertent consequence of z-scoring two hemispheres separately)

No explanation of Fig 5E in the text.

Methods

"This procedure optimally rotates, translates, and scales the right hemisphere" – this doesn't account though for non-linearity of Laplacian embedding which means that that distances across different embeddings are not by default isomorphic. A diagnostic of spectral gaps' similarity across embedding dimensions can be advised to ascertain there are no considerable non-linear distortions for which Procrustes can't compensate. Otherwise, a multi-view embedding is warranted.

"to generate 100 principal gradients for each individual and at the group level" - there is some ambiguity what "group level" means in this context. Embedding of average affinity matrix? Perhaps worth giving a hint of what will be done with individual embeddings in downstream analyses.

"The name of dimensionality reduction, which belongs to the family of graph Laplacians," – I don't think the name was stated.

"The stepwise regression was done using" – I took a note of the author's response, but this is just off the point, except that I now understood what exactly was done using "step-wise" regression. The step-wise regression is almost a misnomer here, at least in a conventional way. Unlike the cases where conventional step-wise regression is typically applied, the gradients are orthogonal by construction, which means the whole procedure can be accomplished by fitting one – 100-gradient - model, and computing unique variance accounted for by each gradient and summing them up as needed. What the authors do is not *wrong*, the result will be equivalent, but the manner in which the result is achieved is redundant/cumbersome and may confuse the reader.

"Adjusted R-squared decreases or is negative when adding more variables into the model in general, indicating collinearity of the model and the need to adjust the entered variables. In the current analysis, we did not find collinearity between gradients" – this is very confusing, mixing apple and oranges. They may decrease even if the predictors are not collinear; for orthogonal variables, this would just signify that a particular variable explains nothing in the data.

Version 2:

Reviewer comments:

Reviewer #1

(Remarks to the Author)

My overall assessment of the manuscript remained largely unchanged over the course of revisions, although the authors have clearly made substantial efforts to temper earlier over-interpretations and to present the results more cautiously, avoiding trivial findings. With these revisions, the study now boils down to: 1) combining two data modalities, fMRI and CMRglc, 2) transforming the fMRI data into a gradient-based representation, 3) applying correlation-/ regression-based analyses to examine relationships between the resulting maps, and 4) demonstrating several forms of association between the two measures.

From a methodological standpoint, the approach does not appear to introduce substantial innovation. As a finding, the reported association is potentially of interest; however, several considerations limit my excitement about this. First, the absolute magnitude of the observed effect seems to be inflated due to the spatial smoothness inherent in the data, such that even randomly permuted maps could explain a nontrivial proportion of the variance. Second, in light of this, it seems plausible that other meaningful brain parcellation - for example, ICA-derived networks - might yield comparable associations. If so, this would suggest that the reported gradient-energy relationship may not be specific to the chosen representation.

COMMSBIO-25-0189-T

Response Letter:

We would like to thank the Editors and Reviewers for their evaluations and constructive comments, and for the opportunity to submit a revised manuscript. Please find our detailed responses below according to the Reviewers' comments. For the revised manuscript, changes are highlighted in yellow and are also listed in the response. Github scripts have been updated and source data are also provided on GitHub. Regarding the figures in the main text, we also provide additional pdf (vector figures) for editing. Revision checklist, editorial checklist, and reporting summary have been uploaded in the system.

Reviewer #1:

Q1: As the length of my recommendations list may suggest, I do not think the paper is strong methodologically. On too many instances the authors report trivial findings and reveal some lack of insight on the methods they use (unless I misunderstood what has been done – the lack of details in Methods section does not help this). The emphasis on looking at the broad range of gradients, in my view, is also problematic methodologically, see details below. It would also be good to get more clarity on whether this association is very specific to gradients. E.g., if, instead of gradients, I would use ICA-derived functional networks (DMN etc), would I fail to account for metabolism in a similar way?

A1: We thank the Reviewer for the evaluation of our work and the critical reflection on our methodological approach. The questions proposed here are specifically mentioned in the comments below so we answer them there. Here we answer the ICA question.

We selected gradients (diffusion map embedding) as our interested variables because it offers a nonlinear continuum (Coifman et al., 2005; Margulies et al., 2016) of low-dimensionality. This approach uses the affinity matrix to preserve local and global structures. The affinity matrix helps in constructing neighborhood graphs for learning lower-dimensional manifolds. It helps us to directly test the hypothesis that similarity of regional energy metabolism is supported by similarity of functional connectivity anchored in continued axes of organization. Therefore, ICA is out of the current scope.

Nevertheless, a previous study compared gradients, parcellation, and sICA to predict behavior (Kong et al., 2023). It suggests that the performance of gradients is higher than sICA. Inspired by geometric eigenmodes (Pang et al., 2023), structural eigenmodes are used to reconstruct the brain activation maps. We used the same stepwise regression model as inputs are functional connectivity gradients (eigenvector) and output is glucose metabolism map. Here, to answer your question, we compared the fitting

using gradients and ICA. When using 100 components, we found the ICA could also predict the FDT-PET map but was less powerful as the fitting curve only starts to increase at 80 components. It indicates that ICA requires a bigger cluster number to converge and the simple solution, e.g., 10 clusters, doesn't capture much of the energy metabolism variance.

We have added this figure as a Supplementary Figure. “**Figure S11. Comparisons between using gradient and independent components, supplementary to Fig. 2B.**”

Q2: Introduction

The first paragraph – I felt as if the authors speak their own language here. I think I understood what they meant (apparently, contrasting local functional characteristics vs long-distance connectivity), but the message here is not transparent if not convoluted. E.g., the measures such as region’s centrality is a characteristic derived based on connectivity (ie., it can also be viewed as “an outcome of inter-regional connectivity” or “topological organization”), hence, the distinction between the two views is not clear. The meaning of “systematically organized by network community integration and segregation” is also non-transparent.

A2: We thank the Reviewer’s comment. Both previous techniques and our approach are using connectivity but the difference is how we summarize it. Graph-based analysis directly binarizes or weights the values then sum the values up. However, gradients calculate the inter-regional connectivity similarity (i.e., affinity matrix), then reduce the dimensionality. If one region’s FC profile is similar to another region’s FC profile, they will share the inter-regional connectivity. The topological organization captured by previous studies (e.g. centrality and local connectivity) reflects node-based connection understanding of the spatial distribution. Rather than asking 'how connected is a region', gradients ask 'where does this region sit along the brain's major organizational axes. We have rephrased this term as “an outcome of inter-regional connectivity profile” to “as an outcome of inter-regional connectome similarity networks”. Higher similarity means that two regions are integrated together and lower similarity means segregation. We have now given the definition in the Methods section (also see below).

Intro: (from Line 42)

“Previous studies have correlated cortical energy cost with regional features of the resting state functional connectome from graph analysis and found that regions with higher local connectivity, centrality, and amplitude of low frequency fluctuations consume more energy⁷⁻¹². Although these indices provide local or network graph information about connectivity patterns, these models only partially explain the spatial pattern of energy expenditure. Another way to understand brain function is to conceptualize it as an outcome of interacting networks on a broader scale or inter-regional connectome similarity networks. Regions with similar functional connectivity (FC) profiles are systematically situated together, which can be captured by low-dimensional embedding gradients¹³. The framework captures the local/global structure of the FC matrix and reflects both integration and segregation of functional networks.”

Methods: (from Line 426)

“Regions with minimal connectivity are similarly farther apart (segregation). Regions having similar connectivity profiles are embedded together along the eigenvector (integration).”

Q3: “Functional gradients are reproducible..” - of relevance for the further analyses : only the first two/three or generally speaking? “We hypothesize that there is a relationship between topological organization of intrinsic functional connections and energy expenditure, as 1) the organization of the cortex is likely a result of evolutionary pressures on minimizing energy demands 8,30,31 and 2) regions that have a similar functional organization likely have a similar energy expenditure 13. “ –not sure that this is well motivated, I don’t think Ref 13 is a ref that demonstrates this. 3) “not how much energy functional organization consumes” – not sure what functional organisation’s energy consumption means.

A3: “Functional gradients are reproducible...”. Yes, previous studies mostly focus on the first two gradients using test-retest data. Now we have revised it to “The first two functional gradients are reproducible...”.

We have reorganized our theoretical framework around three key principles: 1. Metabolic constraint principle: Brain organization reflects evolutionary pressure to minimize energy costs while maintaining function (refs 8,41,42). 2. Functional similarity principle: Regions with similar roles in information processing should have similar metabolic demands - this is what gradients capture through their embedding of functional similarity. 3. Hierarchical organization principle: Brain function is organized along continuous gradients rather than discrete networks, and metabolic demands should reflect this hierarchical structure.

Regarding "functional organization's energy consumption" - we meant to distinguish between the energy cost of maintaining functional connections versus the energy cost of actual neural activity. Due to the extended revision of this version, this sentence has been removed.

Please see below the revision we have made.

from Line 80

“However, with [¹⁸F]FDG-PET, we can only obtain a map of overall energy expenditure within a region. Summarizing the functional connectome matrix to match one [¹⁸F]FDG-PET map is a challenge. To overcome this challenge, and inspired by the structural eigenmodes constraining brain function⁴⁰, we model energy expenditure using

*gradients of functional organization as independent variables. By evaluating the combined variance explained by each gradient, we reconstructed the regional pattern of energy expenditure, illustrating how different gradients contribute to its spatial organization. **Fig. 1** illustrates the principles that regions with similar FC profiles should show similar energy expenditure. The low-dimensionality of the FC similarity matrix (i.e., affinity matrix) was then used to reconstruct the [¹⁸F]FDG-PET map.”*

“Based on this rationale, we tested two specific hypotheses: (1) functional connectivity gradients can reconstruct regional energy expenditure patterns, with model performance improving as more gradients are included; and (2) gradient asymmetry can predict metabolic asymmetry between hemispheres.”

Q4: Figure 1 – I found it very confusing, seems to be a mixture of preprocessing and reasoning blocks. 1) in upper right plane – looks very *grounded* but what does it have in common with the analyses? Are you going to reconstruct all these ATP, ADP, O2 etc? I recon an FDG-PET image would be more fitting here. 2) Equation is likely to be lacking a term for the intercept. The equation seems to be equivalent to “Reconstruct?”, why they are not connected then? 3) I don’t understand what “Regions A and B share etc” box does in between “similar” and “decompose”

A4: We thank the Reviewer for identifying these structural issues with Figure 1. 1) You are absolutely right that the ATP reaction detail was unnecessary for our analysis. We have replaced it with an FDG-PET map that directly shows what we're measuring. 2) We have added the missing intercept term and connected the "Reconstruct?" box directly to the equation to show they represent the same process. 3) We have reorganized the figure to separate preprocessing steps (similarity matrix construction) from the analytical reasoning (gradient decomposition → reconstruction), making the logical flow clearer. The revised figure now follows a clearer sequence: FC profiles → similarity assessment → gradient decomposition → metabolic reconstruction.

Fig. 1: The proposed relationship between glucose metabolism and functional organization.

Q5: “The first aim of our study.. “ – I expected to find a second aim somewhere further in the text, but this was not the case.

A5: We apologise for the omission. We have now included the second aim, which we previously had only written later in the results, which is for asymmetry in the last paragraph of the Main section.

from Line 59

“The second aim of our study was to test this theory using our energy-gradient model in different hemispheres, under the assumption that the spatial pattern of asymmetry in functional brain organization relates to the asymmetry in glucose metabolism.”

Q6: “We hypothesize that when using gradient maps” – it’s the third time throughout the Into when the authors hypothesise something, perhaps the authors need to review this.

A6: As the hypotheses were scattered throughout the introduction before, we have now consolidated all hypotheses into a single, focused paragraph at the end of the introduction: *“Based on this rationale, we tested two specific hypotheses: (1) functional connectivity gradients can reconstruct regional energy expenditure patterns, with model performance improving as more gradients are included; and (2) gradient asymmetry can predict metabolic asymmetry between hemispheres.”* This replaces the three separate hypothesis

statements that were previously scattered throughout the text, providing clearer study aims and better logical flow.

Q7: “the model performance should increase step by step with an increasingly detailed description of functional brain organization (e.g. increased number of gradients and variance explained).” – it will be true even if ones starts adding “random” predictors, I guess the authors mean that it will increase in some statistically meaningful way.

A7: You raise an important statistical point. We address this in several ways: 1) We use adjusted R^2 rather than R^2 , which penalizes for additional variables and prevents spurious improvements from simply adding more predictors. 2) We compare our results against null models where gradient values were randomly shuffled (n=1000 permutations), showing our true predictions exceed 99.4% of null predictions with large effect size (Cohen's $d = 2.42$). 3) Most importantly, we find that improvement plateaus after ~60 gradients, indicating we're capturing meaningful functional organization rather than just adding noise. If improvements were purely due to additional variables, we would expect linear increases without plateaus. This demonstrates that the stepwise improvements reflect genuine functional-metabolic relationships rather than statistical artifacts.

Q8: “under the assumption that the spatial pattern of asymmetry” – is this an assumption or a hypothesis that you want to test using gradients as IVs?

A8: This is an assumption. We have revised this to: "*The second aim of our study was to test this theory using our energy-gradient model in different hemispheres, under the assumption that the spatial pattern of asymmetry in functional brain organization relates to the asymmetry in glucose metabolism.*" (from Line 59)

Q9: Methods.

The method section needs to considerably expanded to include key details. With them lacking, I am not able to provide a coherent recommendation for this section. However, certain processing choices seem to me questionable or at least need a better justification/explanation. Here are some points.

A9: We thank the Reviewer for the suggestions, please see our detailed response below.

Q10: “The images were motion-corrected, high-pass (0.01-0.1 Hz) filtered, regressed out GM, WM, and CSF signals” - does this represent the exact order of preprocessing stages, and if yes this is a bit unorthodox. What does motion-corrected exactly means (e.g., how the motion regressors were constructed). Why PET images could not be registered via native surface templates as it was done to fMRI data, implying

differences in (accuracy of) registration to the template space and double interpolation in the volumetric space for PET data? If there were native surface spaces and tissues segmentations available, I presume there exist structural volumes for each subject. How the mapping between PET/fMRI and structural data was handled? Why different smoothing kernels were chosen to be different between fMRI and PET data? Why was a large (10 mm) smoothing kernel needed for functional data? I believe this will be of a direct relevance for the results, as this will induce spatial dependencies and therefore induce gradient-like features in the data.

A10: Thanks for the comment. We have now preprocessed the fMRI data again (consistent with PET preprocessing steps), which does not change our results (only changes in a few decimal places).

Processing order: The exact order of preprocessing stages is slice-time-corrected, realigned, motion-corrected, skull-stripped, and registered to the anatomical images. Then nuisance signals were regressed-out (scanner drift, physiological noise, and head motion signals), and the time series were band-pass-filtered (0.01 - 0.1 Hz). Finally the images were smoothed (FWHM = 6 mm) and registered to MNI space.

Motion correction: Motion regressors: derived from rigid-body realignment during preprocessing including three translational (X, Y, Z) and three rotational (pitch, yaw, roll) movements.

PET registration: Now the PET images were registered to T1 native space like fMRI preprocessing.

Mapping between native structure and fMRI/PET: Yes, there were native space images. Both fMRI and PET images were aligned to native space using AFNI, which performs a 12-parameter affine transformation including 3 translations (shifts along x, y, z axes), 3 rotations (rotations around x, y, z axes - pitch, roll, yaw), 3 scales (stretching/shrinking along x, y, z axes), 3 shears (skewing the image).

Smoothing kernel: Both fMRI and PET used 6mm in the revised version now. They have no apparent difference between using 6mm or 10mm for the results.

The revised text in Methods:
from Line 374

“Imaging data were acquired on an integrated PET/MR (3T) Siemens Biograph mMR scanner (Siemens, Erlangen, Germany) and used a 12-channel phase-array head coil for the MRI acquisition.

Regarding the PET data collection, they were collected in list-mode format with an average intravenous bolus injection of 184 MBq (SD = 12 MBq) of [¹⁸F]FDG. In parallel to the PET measurement, automatic arterial blood samples were taken from the radial artery every second to measure blood radioactivity using a Twilite blood sampler (Swisstrace, Zurich, Switzerland). The FDG-PET includes 33 dynamic frames: 10 × 12 s, 8 × 30 s, 8 × 60 s, 2 × 180 s, and 5 × 300 s. The attenuation correction was based on the T1-derived pseudo-CT images. The [¹⁸F]FDG-PET images were motion-corrected, spatially smoothed (FWHM = 6mm), and partial volume corrected using the gray matter (GM), white matter (WM), and cerebrospinal fluid (CSF) masks derived from the T1 images. Arterial blood samples were converted to plasma and modeled as a sum of three exponential functions. The arterial input function was calculated by evaluating this function at the times from the PET dynamic frames. The net uptake rate constant (K_i) was estimated using Patlak's plot based on the last five [¹⁸F]FDG-PET preprocessed images and the arterial input function. The CMR_{glc} map was calculated by multiplying K_i by the plasma glucose concentration and then dividing by the [¹⁸F]FDG lumped constant. Finally, the individual CMR_{glc} map was registered to MNI152NLin6ASym 3-mm template and volume to surface mapped to mid-thickness of fsLR-32k space and summarized into 360 parcels using multimodal parcellation (MMP)⁴³.

The fMRI data were acquired during a 10-min time interval using a single-shot echo planar imaging sequence (300 volumes; 35 slices; repetition time, TR = 2000 ms; echo time, TE = 30 ms; flip angle, FA = 90°; field of view, FOV = 192 × 192 mm²; matrix size = 64 × 64; voxel size = 3 × 3 × 3.6 mm³). Anatomical images were based on a T1-weighted 3D-MPRAGE sequence (256 slices; TR = 2300 ms; TE = 2.98 ms; FA = 9°; FOV = 256 × 240 mm²; matrix size = 256 × 240; voxel size = 1 × 1 × 1 mm³). Functional images were slice-time-corrected, realigned, motion-corrected, skull-stripped, and registered to the anatomical images. Thereafter, the global mean intensity was normalized across the fMRI run, the nuisance signals were regressed-out (scanner drift, physiological noise, and head motion signals), and the time series were band-pass-filtered (0.01–0.1 Hz). The regression of the nuisance signals modeled the scanner drift using quadratic and linear detrending, whereas the physiological noise was modeled using the five principal components with the highest variance from the decomposition of white matter and CSF voxel time series. Motion regressors were derived from rigid-body realignment during preprocessing including three translational (X, Y, Z) and three rotational (pitch, yaw, roll) movements. Then, the functional image was spatially smoothed (Gaussian filter, FWHM = 6 mm) and registered to the MNI152NLin6ASym 3-mm template through the anatomical image. Finally, volume to surface function was used to map the time-series volumetric data to mid-thickness of fsLR-32k space and summarized into 360 parcels using MMP. The functional connectome of each individual was calculated by the Fisher-z transforming the time series correlation matrix.”

Q11: Why parcellation had to used? Does it not directly contradict the concept of “gradient” that presumes there are no regional “borders” in the brain but only smooth transition? I guess this is the reason why we call them “gradients”. Otherwise, I would call it by what it precisely is, a non-linear embedding of regional connectivity features.

A11: We acknowledge the contradiction between discrete parcellation and continuous gradients. We used parcellation for computational tractability (downsampling data from fsLR vertex space to region-based space) and to enable comparison with existing literature, but you're correct that this discretizes what should theoretically be continuous transitions. The "gradients" we observe are therefore more accurately described as non-linear embeddings of regional connectivity features, as you suggest. We have revised our terminology throughout to be more precise about what our analysis actually captures versus the theoretical concept of continuous gradients.

Q12: “time series of signal-to-noise ratio (SNR)” – what’s that? I always thought SNR is a mean divided by standard deviation, what is the timeseries thereof?

A12: Here we mean the time series of BOLD signals. To avoid confusion, we have written it as “time series volumetric data”.

from Line 409

“Finally, volume to surface function was used to map the time-series volumetric data to mid-thickness of fsLR-32k space and summarized into 360 parcels using MMP.”

Q13: I would recommend presenting the key information on the imaging parameters here, not by a reference to some other paper.

A13: Thanks for the suggestion. We have now offered the key imaging parameters. Changes can be seen above A10.

Q14: “the affinity matrix in a normalized angle” –What is it, a normalised angle? Was it a name for a transformation applied to convert matrix of z-transformed correlations into affinity matrix?

A14: Yes, it is also called angular distance. Normalized angle correlation refers to a technique where the correlation between two vectors is calculated by first normalizing the vectors (making them unit vectors) and then measuring the cosine of the angle between them. We have added this information in Methods.

from Line 416

“Normalized angle correlation refers to a technique where the correlation between two vectors is calculated by first normalizing the vectors (making them unit vectors) and then measuring the cosine of the angle between them.”

Q15: “Finally, the comparative individual functional gradients were assessed.” – what does “comparative” mean in this context? More details on Procrustes alignment is required, because it is not clear what exactly has been aligned (it’s not matching spatial x,y,z coordinates, correct?)

A15: “Comparative”, apologies we meant “comparable”. Procrustes approach is used to analyse the distribution of a set of shapes. Procrustes alignment can make a shape compared to another, or a set of shapes is compared to an arbitrarily selected reference shape. It rescales the raw data as well as matching spatial x,y,z coordinates. This is a step for individual analysis and cross hemisphere, population, or species analyses (Hong et al., 2019; Liang et al., 2021; Xu et al., 2020). This approach, in our study, we used for alignment between hemispheres. For each individual, we used their raw gradients to run our framework for each of them. We have made relevant revision:

from Line 418

“Next, we employed nonlinear dimensionality reduction to generate 100 principal gradients for each individual and at the group level. At the group level, we set the left hemisphere (LHLH) connectivity gradients as the reference template and aligned the right hemisphere (RHRH) connectivity gradients to this template using Procrustes rotations. This procedure optimally rotates, translates, and scales the right hemisphere gradient space to minimize differences with the left hemisphere template, ensuring that corresponding gradients across hemispheres represent comparable organizational patterns rather than arbitrary mathematical dimensions.”

Q16: “suggested to be relatively robust to noise in the covariance matrix.” – how relevant is this? You have a correlation matrix, obtained after heavy temporal and spatial filtering and averaging across vertices within each parcel. What noise are you referring to here?

A16: Noise in the affinity matrix. We have replaced the “covariance matrix” to “affinity matrix”. This is the parameter (α) for diffusion map embedding. When $\alpha = 0$, the method emphasizes local neighborhood relationships (maximal influence of sampling density), while $\alpha = 1$ focuses purely on global structure (no density influence). The commonly used value of $\alpha = 0.5$ balances these influences, approximating Fokker-Planck diffusion and preserving global relationships between brain regions while maintaining robustness to local

irregularities in the affinity matrix. In the gradient literature as we suggested in the paper, the anisotropic diffusion hyper-parameter is commonly set to $\alpha = 0.5$, a choice that retains global relations between data points in the embedded space (Vos de Wael et al., 2020).

Even after extensive preprocessing (temporal filtering, spatial smoothing, and parcellation), the connectivity affinity matrix may contain residual noise from measurement artifacts, individual variability, and systematic biases. The $\alpha = 0.5$ parameter choice helps ensure that the resulting gradients capture meaningful large-scale organizational patterns rather than being overly influenced by these local irregularities in connectivity estimates.

A17: Stepwise regression – not sure why the emphasis is on stepwise regression. Since the gradients are mutually orthogonal, the authors could just compute the full model, calculate the unique variance per each term and their cumulative sum up to a required component order.

A17: Yes, while the eigenvectors returned by the diffusion map are mutually orthogonal, the gradients exhibit hierarchical functional dependence - each gradient captures variance in the data that remains after considering all previous gradients. This reflects the fundamental nature of dimensionality reduction: gradient 1 explains the most variance, gradient 2 explains the most remaining variance, and so forth.

Therefore, we believe that stepwise regression (G1, then G1+G2, then G1+G2+G3, etc.) is the appropriate approach because: (1) it respects the hierarchical ordering inherent to the dimensionality reduction technique; (2) it reveals the convergence point where additional gradients provide diminishing returns (here, ~60 gradients); and (3) it shows how representation quality improves with each additional gradient component.

Simply summing unique variances would ignore this hierarchical structure and fail to capture how the gradients work together as an ordered set to reconstruct the original connectivity patterns. This stepwise approach aligns with established practices in gradient analysis (Pang et al., 2023) and provides meaningful insight into the dimensionality of cortical organization.

Q18: (After reading the Results section where some addition details of analyses could be found, I have a further comment to the last point. I suspect non-orthogonal gradients can indeed emerge as a result of averaging across subjects (provided this was the case, I hope I understood this correctly). But then how were the gradients matched across subjects. Don't you trivially confuse the gradient order with gradient identity? As an example: assume that in one subject the order of the first (unimodal-transmodal) and the second (sensorimotor-visual) canonical gradients was flipped in one subject but not the others, would you still continue with averaging across subjects without

re-ordering (I hope not). As the gradient order increases, this will for certain lead to decreasing across-subject consistency, and therefore an average of a high-order gradient represents a random mixture.)

A18: As we described above, the gradients can be matched across subjects/hemispheres using Procrustes alignment, which then makes sure that gradient order is also matched. The group-level gradients are computed by the group-level FC matrix across subjects, not “mean gradients”. In the revised individual-level analysis (in the supplements), we used raw gradients to test whether our general concept (regions with similar FC profiles have similar glucose expenditure) can work individually. All individuals can reflect the same fitting trend but different final performance: adjusted R² ranging from 32% to 58% (raw gradients). We also discussed the individual difference, see below:

from Line 315

“Although we had an excellent group-level result, there was still large individual variation (the adjusted R²: 32% to 58%). The reasons might be explained by individual differences in glucose usage types and lack of geometric distance constraints. However, direct biological evidence would be needed to establish that perspective. Geometric distance might constrain the FC⁴⁰. Gaining a functional distance⁵¹ might then improve individual prediction. However, individual data quality and heterogeneity might also play a role.”

Q19: “Adjusted R-squared decreases or is negative when adding more gradients.” – this certainly is not universally true.

A19: No, it is not true for adding more gradients. We intended to say it in a general sense to motivate us to test the correlations between gradients. Because we can not take it as granted before testing the hypothesis that adjusted R² should increase with adding more gradients. Now we revised this sentence:

from Line 448

“Adjusted R-squared decreases or is negative when adding more variables into the model in general, indicating collinearity of the model and the need to adjust the entered variables. In the current analysis, we did not find collinearity between gradients and the Adjusted R-squared stepwise regression displayed the trend of rising first and then converging.”

Q20: “(i.e., 360 for MMP and 8856 for fsLR-5k)” – I got confused. I thought all analyses were done using Glasser parcellation (or some weren’t?)

A20: Yes. Only Glasser parcellation was used in our main results.

Q21: “We manually tuned the penalty parameter from 0.00005 to 0.0003 to visualize..” – what is the criteria for manual tuning. Was it done on subject-by-subject basis?

A21: This tuning was for the asymmetry part, which we have now removed because of the confusion for asymmetry modeling suggested by another Reviewer. Instead, we explained the penalty parameter from 0.05 to 0.3 for the whole cortex modeling in Fig 3B. We tuned this parameter based on the group-level modeling. First, we found when the penalty parameter is greater than 0.03, no gradients can be selected; when the penalty parameter is less than 0.05, all gradients can be selected. We visualize the 6 parameters between 0.05 to 0.3 to demonstrate the regularization process. Here we can dynamically observe which gradients are removed during the process. Please see our changes below:

from Line 452

“In addition, we regularized the step 100 model using balanced lasso and ridge algorithms. We manually tuned the penalty parameter (α) from 0.05 to 0.3 to visualize which gradients can be regularized and how much it can influence the model fit. The range of α values was chosen because no gradients could be selected with α value greater than 0.3 and no gradients could be unselected with α value lower than 0.05.”

Q22: “z-scored LH and RH” – z-score calculated across subjects or a map? “then aligned RH to LH gradients to make them comparable in the main figures” – what is the procedure for the alignment? (looks like more details were presented in Results, but it needs to be more explicit here)

A22: For the group-level analysis, we obtained gradients of the group mean FC matrix. For RH and LH-z-scoring was calculated across the group-level map, not individual-based. We also regard each individual as an independent sample to generalize our model. Then z-scoring was done to each individual and tested the stepwise regression for each individual separately. We have now revised the text:

from Line 457

“The stepwise regression was done using the group-level gradients and CMRglc map, and further generalized for each individual separately. The group-level gradients were calculated from diffusion map embedding on the group mean FC matrix. This group-level and individual-level analysis framework was also employed in asymmetry findings.”

Q23:

Results

“There is no intercorrelation among these gradients (Supplementary Figure S2), indicating the statistical independence of gradients.” – See my comment for stepwise

regression above. My understanding is that gradients represent spatial weights on eigenvectors of a Laplacian matrix, i.e., they are orthogonal by construction. However, if the authors are worried that non-independence could emerge from averaging gradients across subjects, then I don't understand how the authors managed to match them across subjects, as it cannot be assumed that the gradients with the order number N in two subjects are *same*.

A23: As we answered above, group-level gradients were not averaging subjects but diffusion map embedding on the mean FC matrix averaged across subjects. Our motivations to use stepwise regression are 1) gradients are low-dimensional loadings, which means that they follow the one by one order of reduction technique; 2) stepwise regression can offer the perspective “from which number of gradients the fitting curve converges” (here we observed 60 gradients); 3) similar approach has also been used in a previous paper (Pang et al., 2023), where they used eigenmode of brain structure to fit the task fMRI maps using stepwise regression model.

Q24: “to determine whether the regional variance of FC and CMRglc is symmetrically explained by the gradients” – this needs to be better motivated, why it is important to look at FC variance?

A24: That's the core reason why we used stepwise regression and answered above. We want to observe the detailed procedures. If the FC variance and PET variance can be synchronized explained by the number of gradients, it could be a strong factor to support our hypothesis: regional difference of glucose expenditure is captured by the similarity of their functional connectivity profiles.

Q25: “we used slope and mean absolute error” – what does mean absolute error tell us?

A25: Mean absolute error tells us more about the loss: how close predictions are to the ground truth. Slope is more telling how fast the x changes with y. They can give us different information. We have added this information in the manuscript:

from Line 165

“Slopes close to 1 indicate faster synchronization between the FC matrix and CMRglc map via gradients. Smaller MAE values indicate closer predictions to the ground truth.”

Q26: “manually adjusted penalty parameters” – I think “manually” is confusing in this context. If I understood this correctly, the authors computed regularised regression for a range of penalty values, what is “manual” about it?

A26: Thanks for the comment. “Manually” might cause a lot of confusion. Now we revised it as “for a range of penalty values”.

Q27: “**With increasing penalty parameters, the Pearson r between true CMRglc and model-predicted CMRglc was reduced**” – of course it will be, as the heavier penalty will reduce an effective degree-of-freedom of the fitting model. I guess it would be better if - instead of reporting this - the authors work out quantitatively the optimal penalty. “**This illustrates that low dimensions that explain a lot of variance..**” the sentence contains both interesting and trivial findings. The fact that the strongest (spatially distributed) gradients account for the metabolism is interesting. The fact that weakest gradients account less for metabolism is most likely to be trivial. I presume, as the order number of a gradient increases, their spatial composition becomes more and more local, in effect, accounting for the local deviations in metabolism (the relationship which potentially can be induced by both intrinsic and applied smoothing in the data).

A27: Thank you for your comment. Our fitting curve and adjusted R^2 show that all gradients contribute meaningfully to the model - none are redundant. The optimal model uses all gradients without penalty. We used the penalty analysis to identify which gradients are most important as constraints increase, as described in our manuscript. This approach supports our finding that "low dimensionalities explain a lot of variance" and demonstrates that early gradients carry more information than later ones. We agree with your observation that "as the order number of a gradient increases, their spatial composition becomes more and more local." We've added this point to our discussion (see changes below). Regarding your suggestion that diminishing metabolic relevance of higher-order gradients might reflect methodological factors like smoothing. We don't think this significantly affects our results. We tested different smoothing parameters for fMRI data (10mm in the previous version vs 6mm in the current version) and found consistent results - both versions converge at around 60 gradients with similar performance. We initially considered intrinsic smoothing as a potential confound, but our spatial autocorrelation analysis in Fig. 4 suggests otherwise. The true map consistently outperforms spatially permuted versions, indicating our findings aren't simply due to smoothing artifacts.

from Line 307

“Our model, which uses the topography of intrinsic brain function, achieved a stronger relationship. As the gradient number increased (e.g., higher-order latent dimensions), their spatial composition became more local.”

Q28: “**variogram**” – I think some description in Methods is needed of what is achieved by variograms, e.g., is it a random rotation of gradients or something else?

A28: We are happy to briefly describe it here. A variogram quantifies spatial autocorrelation (using geometric distance) to compare spatial dependencies along different orientations. In our case, we used variogram to obtain the surrogate maps of null gradients considering spatial autocorrelation, which was done with the help of inter-regional geometric distance. Detailed methodology was offered by the paper (Burt et al., 2020). We used this approach for both gradients and FDT-PET maps separately.

from Line 206

“Considering spatial autocorrelation^{36,47-49} of brain maps, we used a variogram to permute the brain maps based on the geometric distance matrix of central coordinates of the Glasser atlas. Therefore, it can generate spatially dependent surrogate maps on the target map.”

Q29: “there was a strong correlation ($r = 0.732$ with 100 gradients)” – between reconstructed and actual map?

A29: Yes, it was. Because we have now re-organized this part, these findings are not included anymore.

Q30: I am not sure I understand accurately what has been done in the “hemispheric independence” section. Here are a few points: “we found that the variance in energy asymmetry can be explained by the combined asymmetry of gradients” – Is this a meaningful result, unlikely to be matched by random gradients with matched spatial correlation properties?

A30: Thanks for pointing this out! We removed the hemispheric independence section and integrated those results into our 5-mode asymmetry analysis. This decision followed careful consideration of the trade-off between reduced adjusted R^2 and increased correlation in our previous asymmetry modeling.

Regarding spatial properties, while there may be some influence, our current focus is comparing the 5 modes rather than examining spatial autocorrelations within each mode.

The 5 modes primarily capture hemispheric competition and lateralization by testing whether the two hemispheres share energy regression coefficients. They are conceptually distinct:

- Lateralization is defined by hemispheric domains, where energy sources are allocated to the "winning" hemisphere
- Competition is defined by hemispheric cooperation, where energy sources are allocated dynamically between hemispheres

These represent two different perspectives for understanding brain function. The competition model offers a novel viewpoint that we hope will enhance diversity in brain asymmetry research, though the neurobiological meaning still requires future investigation.

The updated description for the 5 modes is:

from Line 233

*“We tested whether left and right hemispheres exhibit distinct relationships between functional organization and glucose metabolism (hemisphere-specific models, modes 1-2) or whether they share common organizational principles that can be applied across hemispheres (shared models, modes 3-5). The hemisphere-specific modes reflect LH and RH separately for ‘hemispheric competition’. The shared modes indicate ‘hemispheric lateralization’. Hemispheric lateralization reflects intrinsic functional difference between the two hemispheres that exists without difference of energy relocation. Hemispheric competition reflects functional difference between two hemispheres relies on different energy demands, so two hemispheres don’t share the energy-function coefficients. Then, we compared the competition and lateralization modes in **Fig. 5F**. We used covariance and slope instead of correlation to indicate the closeness between predicted and true energy asymmetry maps to constrain both within one scale. Mode predicted brain maps are shown in **Supplementary Fig. S4**. The covariance scores between true and predicted CMRglc asymmetry maps for these modes in step 100 were 0.058, 0.041, 0.006, and 0.011, and 0.051. The slopes of mode 1 to 5 were: 0.603, 0.424, 0.065, 0.113, 0.530. This suggests the competition model performs better than the lateralization model even with cross-hemispheric optimizing parameters.”*

Q31: “To address this, “ – to address what? Rectifying the relation between the order and predictive power? “energy-guided gradient asymmetry maps” – not sure I understand what it is.

A31: Thanks for the comment. As we answered above, we have deleted this section and compared the five modes together now.

Q32: “but slightly worse when predicting the other hemisphere..” – unless I misunderstood this – if this is the difference in the variance accounted for in training and testing samples (=hemispheres), then this is the most trivial result.

A32: Yes. We have deleted this trivial result for clarity.

Q32: “LH and RH use independent training coefficients” – is this fitted regression coefficients? (please clarify)

A32: Yes, this is fitted regression coefficients. We trained one hemisphere and fit the same hemisphere using the trained regression coefficients, so we called it independent training coefficients. We have now changed the wording of training coefficients to regression coefficients.

Q33: Paragraph starting with “To further determine whether LH and RH have” - My understanding of the procedure is that the authors fitted LH and RH to corresponding LH and RH energy maps. They then compared which of 4 combinations of IVs and DVs and one asymmetry fit is the best-fitting. However, I am missing a logical link on how this is informative of “competition” vs “lateralisation”. The result is clearly expected based on the results reported in the above (Figure 5C vs Figure 6A).

A33: The core definition for them is whether energy optimization for left and right hemispheres is dependent or independent (“sharing optimization” in Figure 5B). If independent (LH and RH don’t share regression coefficients, modes 1 and 2), then it’s competition. If dependent (LH and RH share the same regression coefficients, modes 3, 4, and 5), it is lateralization. The equation for best lateralization has been given in Figure 5E. The concept of brain lateralization is that one hemisphere dominates a function and the other hemisphere is less involved in this function. The reflection in the energy-function relationship is that functional difference between two hemispheres is intrinsic and exists without difference of energy relocation. Brain competition reflects functional difference between two hemispheres relies on different energy demands, so two hemispheres don’t share the energy-function coefficients.

We have revised the information. See below:

from Line 233

“We tested whether left and right hemispheres exhibit distinct relationships between functional organization and glucose metabolism (hemisphere-specific models, modes 1-2) or whether they share common organizational principles that can be applied across hemispheres (shared models, modes 3-5). The hemisphere-specific modes reflect LH and RH separately for ‘hemispheric competition’. The shared modes indicate ‘hemispheric lateralization’. Hemispheric lateralization reflects intrinsic functional difference between the two hemispheres that exists without difference of energy relocation. Hemispheric competition reflects functional difference between two hemispheres relies on different energy demands, so two hemispheres don’t share the energy-function coefficients.”

Trainee Reviewers comments:

Q1: Whilst I feel that the overall premise of comparing BOLD response to energy consumption in the brain through the use of rfMRI and PET imaging is an interesting one, the work within this paper doesn't quite fulfil the potential that you intended it to.

Main:

As mentioned above the language used throughout the paper is somewhat colloquial. The flow of this section is disjointed and jumps backwards and forwards. Throughout this section you mention three different hypotheses may be good group together after your main background information.

A1: We have re-organized the section to improve clarity.

from Line 89

“Based on this rationale, we tested two specific hypotheses: (1) functional connectivity gradients can reconstruct regional energy expenditure patterns, with model performance improving as more gradients are included; and (2) gradient asymmetry can predict metabolic asymmetry between hemispheres. With regards to the first hypothesis, although generally only the first three gradients have been evaluated and functionally interpreted^{35,41}, we extended this to a more exhaustive model and observed how the model fits with the first 100 gradients. This decision was motivated by previous work that showed prediction accuracy of behavior improved by including 60 gradients or more⁴². The variance explained by these gradients may capture brain features with behavioral and physiological relevance⁴². Moreover, it will allow us to understand whether energy patterning explained by the gradient model gradually increases with the number of gradients, or reaches a ceiling after a few dominant organizational principles. For the second hypothesis, we assessed asymmetry in functional gradients by training hemisphere-specific parameters. Gradient asymmetry could then be used to predict asymmetry in glucose metabolism, either directly or by training separate models for each hemisphere.”

Q2: On page 4, it is noted what your first aim was, however, there are no following aims after this point. Did you intend to have further aims for this work? Or was it the intention to only have one? If the later, I would consider rewording this so that you only mention that have one aim. Additionally, it would be good to separate this out from your ‘Main’ section to make it clear that this is the aim of your work.

A2: Yes, we indeed have two aims. Now we have revised the “aims” section in the first paragraph, please see below.

from Line 52

“The primary aim of this work was to study the relationships between energy consumption and organization gradients of intrinsic brain function. Moreover, both functional organization and glucose metabolism show hemispheric differences¹⁴⁻²¹.”

from Line 59

“The second aim of our study was to test this theory using our energy-gradient model in different hemispheres, under the assumption that the spatial pattern of asymmetry in functional brain organization relates to the asymmetry in glucose metabolism.”

Q3: Throughout this section the references (1-6, 14-16 and 20,21) that you use are 12+ years old, there are more recent references available to support your points here.

A3: They are old but we think it’s appropriate to discuss the fMRI-PET modeling history or basis. Other references such as 7-12 (they are more recent). We kept more than one third of the references in our manuscript for the last 5 years.

Q4: Methods/Results

The first paragraph of your results section is a very limited methodology of how you have conducted this study. The small paragraph stating you are using a secondary data source in the results section could be used within a methods section, adding total number of data sets analysed, with median and age ranges of the participants, you can then go onto say that this was subdivided into experimental and replication datasets. It would also be worth clarifying what differences there were between these subsets are if any.

A4: We are happy to clarify the writing formats for Nature-series journals. In general, little methodology information is given in the Results section and detailed (no repetitive) methodology should be given in the Methods section. Nature-series journals usually put the Methods section after Discussion. Regarding “subdivided into experimental and replication datasets”, because we wanted to increase the quality of the data, we did group-level analysis first (like we need to generate a template), which requires averaging metrics across as many people as we can. We also replicated the analysis for each individual instead of the group-level analysis averaging either experimental or replication datasets. We think this way is to observe the individual differences, which is better than the subset differences. The individual replication has been displayed in our supplementary figures .

Q5: Whilst I see that you have put a large methods section at the end of the paper, it would be beneficial and better placed to have a well described methods section prior to

the results section, to limit confusion of what your method was throughout the paper. As this leaves the reader unclear throughout your results section about how you have carried out your analysis of the secondary data set. More of this final methods section should be condensed into a methods section prior to the results section.

A5: We really appreciate your suggestions. As we answered above, however, it's Nature-series journals classical formatting. We have now enlarged the Methods section with more imaging parameters, preprocessing details, and analysis steps, and interpretation of methodology. We hope these changes can be helpful. Please see the revision in A7.

Q6: Additionally, within the methods section it would be beneficial to ascertain when the data was collected and over what time period.

A6: We have now added the relevant information. *“The data were collected in the mornings using the same scanning parameters for both datasets. The experimental dataset was collected between February 2016 and September 2017, whereas the replication dataset was acquired between February and March 2021.”*

Q7: Parameters are mentioned as part of the initial paper for the secondary paper, are the parameters stated for fMRI or for the PET imaging too? Currently this is unclear to what parameters are noted within this original publication.

A7: Yes, the parameters are stated for fMRI or for the PET imaging too because we directly used the data from “the secondary paper”. We have now added the detailed imaging parameters in the Methods section. Please also see below:

from Line 374

“Imaging data were acquired on an integrated PET/MR (3T) Siemens Biograph mMR scanner (Siemens, Erlangen, Germany) and used a 12-channel phase-array head coil for the MRI acquisition.

Regarding the PET data collection, they were collected in list-mode format with an average intravenous bolus injection of 184 MBq (SD = 12 MBq) of [18F]FDG. In parallel to the PET measurement, automatic arterial blood samples were taken from the radial artery every second to measure blood radioactivity using a Twilite blood sampler (Swisstrace, Zurich, Switzerland). The FDG-PET includes 33 dynamic frames: 10 × 12 s, 8 × 30 s, 8 × 60 s, 2 × 180 s, and 5 × 300 s. The attenuation correction was based on the T1-derived pseudo-CT images. The [18F]FDG-PET images were motion-corrected, spatially smoothed (FWHM = 6mm), and partial volume corrected using the gray matter (GM), white matter (WM), and cerebrospinal fluid (CSF) masks derived from the T1 images. Arterial blood samples were converted to plasma and modeled as a sum of three exponential functions. The arterial input

function was calculated by evaluating this function at the times from the PET dynamic frames. The net uptake rate constant (K_i) was estimated using Patlak's plot based on the last five [^{18}F]FDG-PET preprocessed images and the arterial input function. The CMR_{glc} map was calculated by multiplying K_i by the plasma glucose concentration and then dividing by the [^{18}F]FDG lumped constant. Finally, the individual CMR_{glc} map was registered to MNI152NLin6ASym 3-mm template and volume to surface mapped to mid-thickness of fsLR-32k space and summarized into 360 parcels using multimodal parcellation (MMP) ⁴³.

The fMRI data were acquired during a 10-min time interval using a single-shot echo planar imaging sequence (300 volumes; 35 slices; repetition time, $\text{TR} = 2000$ ms; echo time, $\text{TE} = 30$ ms; flip angle, $\text{FA} = 90^\circ$; field of view, $\text{FOV} = 192 \times 192$ mm²; matrix size = 64×64 ; voxel size = $3 \times 3 \times 3.6$ mm³). Anatomical images were based on a T1-weighted 3D-MPRAGE sequence (256 slices; $\text{TR} = 2300$ ms; $\text{TE} = 2.98$ ms; $\text{FA} = 9^\circ$; $\text{FOV} = 256 \times 240$ mm²; matrix size = 256×240 ; voxel size = $1 \times 1 \times 1$ mm³). Functional images were slice-time-corrected, realigned, motion-corrected, skull-stripped, and registered to the anatomical images. Thereafter, the global mean intensity was normalized across the fMRI run, the nuisance signals were regressed-out (scanner drift, physiological noise, and head motion signals), and the time series were band-pass-filtered (0.01–0.1 Hz). The regression of the nuisance signals modeled the scanner drift using quadratic and linear detrending, whereas the physiological noise was modeled using the five principal components with the highest variance from the decomposition of white matter and CSF voxel time series. Motion regressors were derived from rigid-body realignment during preprocessing including three translational (X, Y, Z) and three rotational (pitch, yaw, roll) movements. Then, the functional image was spatially smoothed (Gaussian filter, $\text{FWHM} = 6$ mm) and registered to the MNI152NLin6ASym 3-mm template through the anatomical image. Finally, volume to surface function was used to map the time-series volumetric data to mid-thickness of fsLR-32k space and summarized into 360 parcels using MMP. The functional connectome of each individual was calculated by the Fisher-z transforming the time series correlation matrix.”

Q8: I do feel that it is good that you have broken down your report into a section beneficial to the reader. Throughout the hemispheric independence section your method is unclear throughout. LH and RH you mention that you have trained the algorithm separately for this, it is unclear throughout this section as to what your method and thought process was.

A8: The hemispheric independence: we trained regression coefficients for RH and LH independently. After obtaining the regression coefficients, we proposed 5 modes to explain the hemispheric modelling of fMRI and PET. The 5 modes please see:

from Line 233

“We tested whether left and right hemispheres exhibit distinct relationships between functional organization and glucose metabolism (hemisphere-specific models, modes 1-2) or whether they share common organizational principles that can be applied across hemispheres (shared models, modes 3-5). The hemisphere-specific modes reflect LH and RH separately for ‘hemispheric competition’. The shared modes indicate ‘hemispheric lateralization’. Hemispheric lateralization reflects intrinsic functional difference between the two hemispheres that exists without difference of energy relocation. Hemispheric competition reflects functional difference between two hemispheres relies on different energy demands, so two hemispheres don’t share the energy-function coefficients. Then, we compared the competition and lateralization modes in Fig. 5F.”

Q9: Discussion

First sentence is clear and you should think about how this is structured and possibly stating a similar statement earlier within the work for clarity or what you are trying to convey. You mention in paragraph one that you are investigating the differences between the hemispheres and if this is related to their energy consumption. Shouldn’t this have been mentioned earlier when mentioning your aims?

A9: We have now added all the aims in the first paragraph of Main. We hope that the first paragraphs of Main and Discussion are structurally matched.

from Line 52

“The primary aim of this work was to study the relationships between energy consumption and organization gradients of intrinsic brain function. Moreover, both functional organization and glucose metabolism show hemispheric differences¹⁴⁻²¹.”

from Line 59

“The second aim of our study was to test this theory using our energy-gradient model in different hemispheres, under the assumption that the spatial pattern of asymmetry in functional brain organization relates to the asymmetry in glucose metabolism.”

Q10: Stepwise Linear Regression Model, do you have a reference for this?

A10: Yes, we have a reference for that, see (Pang et al., 2023). Our motivation to use stepwise linear regression model is 1) gradients are low-dimensional loadings, which means that they follow the one by one order of reduction technique; 2) stepwise linear regression model can offer the perspective “from which number of gradients the fitting curve converges” (here we observed 60 gradients); 3) similar approach has also been used in a

previous paper (Pang et al., 2023), where they used eigenmode of brain structure to fit the task fMRI maps using stepwise linear regression model.

Q11:

Conclusion

Would benefit being in a section on its own to define this, starting with ‘In conclusion...’ or ‘In Summary we found....’

A11: We have now used “In conclusion”.

Q12: Part of your aim in section ‘Main’ is to ‘evaluate to what extent functional organization and energy expenditure have shared organizational features’, you have loosely linked this back to the aim but could be more succinct. Additionally, you mention that this could provide further perspectives for its change in neuropsychiatric and neurological disorders. By perspectives do you think this could instruct change in patient pathways and inform treatment plans?

A12: Thanks for the comment! Why we said it could be applied to neuropsychiatry is because of our individual-level analysis. Even though the group-level reached an excellent fitting, we found that individual performance actually varied quite a bit (ranging from 32% to 58%). This variability may reflect individual differences in brain organization that could be relevant for understanding neuropsychiatric conditions, where altered connectivity-metabolism coupling has been reported. While we do not have patient data in the current study, the framework's ability to capture individual differences in organization-metabolism relationships provides a foundation for future clinical applications.

Q13: Figures

There have been a lot of figures provided to illustrate your work, with supplementary material providing an additional 22 to the figures within the main text. May be beneficial to evaluate what each of the figures are adding to your work and if all are necessary to illustrate your points and what they are adding to the overall message of the work.

A13: Thanks for your comment. We have now removed some supplementary analyses and reduced the number of supplementary figures from 22 to 10.

Reviewer #2:

Q1: Main (or Introduction):

While the main ideas of the study are all present in the introduction, some refactoring is needed. The hypotheses paragraph (3rd paragraph) is too long and lacks a clear, linear flow. The main hypothesis is well-stated, but the transitions between concepts—FDG-PET limitations, gradient-based modeling, and the rationale for testing 100 gradients—are abrupt. The argument would be clearer if broken into distinct sections: (1) Hypothesis and Background, (2) FDG-PET Limitations and Justification for Gradients, and (3) Rationale for Expanding Gradient Use. The sudden shift to testing 100 gradients should be introduced more smoothly, explaining why this is necessary. Improved transitions and a more structured approach would enhance clarity and readability.

The last paragraph related to brain asymmetry should be merged with the previous hypotheses paragraph, and a final concluding paragraph that highlights the main findings should be created to finalize the introduction. Overall, a refactoring of the Main section is needed.

A1: We thank the Reviewer for the constructive comment. We have now revised the introduction to make this clearer. Also together with other Reviewers' comments, we put the hypotheses into one section, integrating the rationale for 100 gradients afterwards, to make the transitions smooth. The rationale for the 100 gradients choice is that a previous paper suggested that prediction accuracy of behavior improved by including 60 gradients or more (Kong et al., 2023). And when we fitted the model with as many gradients as possible until the performance does not increase (first trying 100 gradients). We observed that after 60 gradients, the fitting curve converged. So finally we visualize the whole procedure of 100 gradients. Please see below where we made the changes.

from Line 89

“Based on this rationale, we tested two specific hypotheses: (1) functional connectivity gradients can reconstruct regional energy expenditure patterns, with model performance improving as more gradients are included; and (2) gradient asymmetry can predict metabolic asymmetry between hemispheres. With regards to the first hypothesis, although generally only the first three gradients have been evaluated and functionally interpreted^{35,41}, we extended this to a more exhaustive model and observed how the model fits with the first 100 gradients. This decision was motivated by previous work that showed prediction accuracy of behavior improved by including 60 gradients or more⁴². The variance explained by these gradients may capture brain features with behavioral and physiological relevance⁴². Moreover, it will allow us to understand whether energy patterning explained by the gradient model gradually increases with the number of gradients, or reaches a ceiling after

a few dominant organizational principles. For the second hypothesis, we assessed asymmetry in functional gradients by training hemisphere-specific parameters. Gradient asymmetry could then be used to predict asymmetry in glucose metabolism, either directly or by training separate models for each hemisphere.”

We also revised/further clarified all our aims in the first paragraph. Please see below where we made the changes.

from Line 52

“The primary aim of this work was to study the relationships between energy consumption and organization gradients of intrinsic brain function. Moreover, both functional organization and glucose metabolism show hemispheric differences ¹⁴⁻²¹. The classical theory of why the cortex exhibits functional asymmetry posits that it avoids costly duplication of neural circuits with the same function ²², which forms the dominant hemisphere or lateralization ²³. However, this prediction has been difficult to test because direct measurements for energy cost across the whole cortex were lacking. The second aim of our study was to test this theory using our energy-gradient model in different hemispheres, under the assumption that the spatial pattern of asymmetry in functional brain organization relates to the asymmetry in glucose metabolism.”

Final Main concluding paragraph: (from Line 103)

“We found that combining the first 5 gradients could rival the predictive power of models based on graph topological metrics (Pearson r ranging from 0.41 to 0.51), ultimately bridging global and regional features of brain organization. After 60 gradients, the variance explained converged (Pearson r ranging from 0.86 to 0.88, adjusted R -squared ranging from 70.0% to 72.1%). Through the hemispheric modelling, we observed that the hemispheric difference of combined gradients better interpreted the glucose metabolism, rather than the hemispheric difference of single gradients (ratio = 1.14:1).”

Q2: Results

The results section is detailed and appears to stem from a robust methodology. However, the rationale behind the "Model Explanation and Regularization" section and its underlying methods is unclear. This section “Hemispheric independence” presents a well-structured comparison of hemispheric asymmetry models, demonstrating that hemispheric competition outperforms lateralization in explaining metabolic asymmetry. The clear comparisons between different energy-sharing modes strengthen the conclusion that LH and RH operate more independently than interdependently. However, one notable result—the negative covariance in Mode 2 (cross-hemisphere parameter application)—should be more clearly presented, as it indicates an inverse relationship rather than a simple failure of generalization. The

quantitative comparisons across models are well-executed, but providing additional summary statistics (e.g., variance explained, R²) alongside covariance scores would help better contextualize model performance. These adjustments would enhance the clarity of the results before interpretation in the Discussion section.

A2: We thank the Reviewer for the constructive comments. We have now revised the explanation for Mode 2. Because we did not flip energy asymmetry for that mode but training for LH and RH was crossed, the variance score was negative. Now we flipped asymmetry for mode 2 to correct it. We now also used another index “slope” to compare them instead of Pearson r and R² because we want to avoid scale differences. Where 1 unit change of X is related to 1 unit change of Y. Pearson r and R² will make the score high if they are not in the same scale (e.g., A is in [-0.5,0.5] and B is in [0,1]). Our model used the z-score of input and already forced the predicted value to be on the same scale: [-1,1]. We added the slope information below:

from Line 245

“The covariance scores between true and predicted CMRglc asymmetry maps for these modes in step 100 were 0.058, 0.041, 0.006, and 0.011, and 0.051. The slopes of mode 1 to 5 were: 0.603, 0.424, 0.065, 0.113, 0.530. This suggests the competition model performs better than the lateralization model even with cross-hemispheric optimizing parameters.”

Q3: Discussion

The discussion provides a strong synthesis of the study’s key findings, particularly the robust relationship between functional gradients and energy consumption (~70% variance explained) and the importance of sparsity in functional connectivity for predicting metabolic demand. The results convincingly demonstrate that hemispheric competition better explains metabolic asymmetry than lateralization, challenging the assumption that functional asymmetry alone optimizes brain energy consumption. However, several important findings are not adequately addressed.

First, while the study includes individual-level analyses to complement group-level findings, the results show greater variability and weaker predictive power, particularly for asymmetry modeling (max adjusted R² = 3.3%). The discussion does not acknowledge this limitation or explore why functional-metabolic coupling is less stable across individuals. Addressing whether this is due to inter-individual variability, methodological constraints, or the inherent difficulty of modeling metabolic asymmetry at the individual level would improve the interpretation.

A3: We thank the Reviewer for the constructive comments! Regarding the individual-level analyses, we have further discussed possible reasons causing the individual variation such as

different glucose usage types, FC explained the aerobic glycolysis. While anaerobic glycolysis might happen in astrocytes. Other metrics such as individual geometric distance might constrain the FC, then improve individual prediction. However, individual data quality and heterogeneity might also count as well. Regarding the asymmetry modelling part, we have re-organized the results to avoid confusion (also see Q5). Now the “(max adjusted $R^2 = 3.3\%$)” has been removed and we went directly into different asymmetry modes as well as for individual asymmetry. Our main focus was to study whether asymmetry can be explained by lateralization (20% of individuals) or competition (80% of individuals). We made relevant revisions for interpretation of individual differences. Please see below.

from Line 315

“Although we had an excellent group-level result, there was still large individual variation (the adjusted R^2 : 32% to 58%). The reasons might be explained by individual differences in glucose usage types and lack of geometric distance constraints. However, direct biological evidence would be needed to establish that perspective. Geometric distance might constrain the FC⁴⁰. Gaining a functional distance⁵¹ might then improve individual prediction. However, individual data quality and heterogeneity might also play a role.”

Q4: Second, the unexpected negative covariance in cross-hemispheric predictions (Mode 2) suggests that applying one hemisphere’s functional-metabolic mapping to the other results in systematic “misprediction,” yet this is not discussed. It remains unclear whether this indicates true hemispheric metabolic competition—where energy use in one hemisphere negatively correlates with the other—or if it arises from modeling constraints or alignment issues. Clarifying this result would strengthen the conclusions on hemispheric independence.

A4: Regarding the cross-hemispheric predictions, as we answered above, the negative value is not unexpected because we did not flip CMRglc asymmetry map. We have now corrected it.

from Line 233

“We tested whether left and right hemispheres exhibit distinct relationships between functional organization and glucose metabolism (hemisphere-specific models, modes 1-2) or whether they share common organizational principles that can be applied across hemispheres (shared models, modes 3-5).”

Q5: Third, while the study shows that higher-order gradients do not improve asymmetry modeling, the discussion does not elaborate on why functional asymmetry does not translate into metabolic asymmetry. The results suggest that stronger

functional connectivity patterns contribute more to regional energy demand, yet this relationship appears to break down in the context of asymmetry. A brief reflection on why functional gradients fail to explain metabolic asymmetry—and whether this is due to differences in hemispheric specialization, vascular factors, or the limitations of functional gradients alone—would provide a more complete interpretation. Addressing these points would enhance the clarity and depth of the discussion, ensuring that all key findings are properly contextualized. Finally, the section in this current form does not discuss the limitations and future steps of the study.

A5: Thank you for this feedback. Upon deeper reflection, we've reinterpreted our findings regarding functional and metabolic asymmetry. While the adjusted R^2 wasn't as high as initially expected, the correlation remained strong. Rather than viewing this as a failure, we recognized it likely reflects collinearity between functional asymmetries, a hypothesis confirmed when we examined the correlation matrix between functional asymmetries. To make the asymmetry analysis more accessible, we streamlined our approach by:

1. Removing the standalone asymmetry figure
2. Integrating those results directly into our 5-mode analysis
3. Consolidating to a single figure that presents all asymmetry results

The revised version now presents asymmetry results more clearly through our integrated approach, with interpretations provided as follows:

from Line 343

“This suggests that energy asymmetry may not result solely from cumulative functional lateralization in topology. While asymmetry of functional gradients has been extensively studied ^{15,17,18,20,21}, our results suggest that gradients may not only reflect static brain organization but also how the brain operates across different spatial scales or in the form of spatial competition in the context of energy optimization. Such an interpretation could be in line with conceptualizations that functional lateralization results from interactions between hemispheric functions ^{55,56}.”

This restructuring better captures the relationship between functional and metabolic asymmetry while improving the manuscript's clarity and flow. We also further discussed it in the limitation:

from Line 352

“There are also limitations to this study. First, although we found individual variability, the reasons are certainly more complex than biology and demographics. Future studies could enhance the sample size and add more interesting variables such as genetics (twins) and

psychological factors (mental health). Second, functional connectivity is more stable with more time series obtained such as the Human Connectome Project (1200 time series). Increasing the time series will further help identify the redundancy organization informed by glucose metabolism. Last, regarding the comparisons between lateralization and competition models, we simply applied LH or RH trained parameters. More complex inter-hemispheric models might provide a more comprehensive result.”

Q6: Methods

More details on the elastic net regularization approach should be provided, including the specific implementation, choice of hyperparameters (e.g., α values), and how they were optimized or selected. Further clarification on the parcellation scheme is needed, particularly regarding its resolution, whether it was symmetrically applied across hemispheres, and how hemispheric differences (if any) were accounted for in the analyses. Providing these details would improve the transparency and reproducibility of the methodology.

A6: We tuned this parameter based on the group-level modeling. First, we found when the penalty parameter is greater than 0.03, no gradients can be selected; when the penalty parameter is less than 0.05, all gradients can be selected. So we visualize the 6 parameters between 0.05 to 0.3 to demonstrate the regularization process. Therefore, we can dynamically observe which gradients are removed during the process. For the parcellation scheme, we selected Glasser parcellation (MMP) because it offers homologous regions across hemispheres. We have now revised the Methods section.

from Line 454

“The range of α values was chosen because no gradients could be selected with α value greater than 0.3 and no gradients could be unselected with α value lower than 0.05.”

from Line 462

“To quantify the inter-hemispheric differences, we calculated asymmetry by LH minus RH. MMP offers an atlas of homologous regions between hemispheres allowing asymmetry to be calculated.”

Q7 - General comments:

Refining the language throughout the manuscript would improve clarity and readability. The use of non-scientific terminology should be avoided. For example, in the Methods section, the repeated use of the term "kick out" should be replaced with a more precise term such as "removed," "excluded," or "penalized" depending on the intended meaning.

A7: Thanks for the suggestion. We, as well as an invited English native speaker, have now gone through the entire manuscript and rephrased the non-scientific terms.

References:

- Berardi, A., Haxby, J. V., Grady, C. L., & Rapoport, S. I. (1991). Asymmetries of brain glucose metabolism and memory in the healthy elderly. *Developmental Neuropsychology*, 7(1), 87–97. <https://doi.org/10.1080/87565649109540478>
- Burt, J. B., Helmer, M., Shinn, M., Anticevic, A., & Murray, J. D. (2020). Generative modeling of brain maps with spatial autocorrelation. *NeuroImage*, 220, 117038. <https://doi.org/10.1016/j.neuroimage.2020.117038>
- Castrillon, G., Epp, S., Bose, A., Fraticelli, L., Hechler, A., Belenya, R., Ranft, A., Yakushev, I., Utz, L., Sundar, L., Rauschecker, J. P., Preibisch, C., Kurcyus, K., & Riedl, V. (2023). An energy costly architecture of neuromodulators for human brain evolution and cognition. *Science Advances*, 9(50), eadi7632. <https://doi.org/10.1126/sciadv.adi7632>
- Coifman, R. R., Lafon, S., Lee, A. B., Maggioni, M., Nadler, B., Warner, F., & Zucker, S. W. (2005). Geometric diffusions as a tool for harmonic analysis and structure definition of data: Diffusion maps. *Proceedings of the National Academy of Sciences*, 102(21), 7426–7431. <https://doi.org/10.1073/pnas.0500334102>
- Gonzalez Alam, T. R. del J., Mckeown, B. L. A., Gao, Z., Bernhardt, B., Vos de Wael, R., Margulies, D. S., Smallwood, J., & Jefferies, E. (2022). A tale of two gradients: Differences between the left and right hemispheres predict semantic cognition. *Brain Structure and Function*, 227(2), 631–654. <https://doi.org/10.1007/s00429-021-02374-w>
- Hong, S.-J., Vos de Wael, R., Bethlehem, R. A. I., Lariviere, S., Paquola, C., Valk, S. L., Milham, M. P., Di Martino, A., Margulies, D. S., Smallwood, J., & Bernhardt, B. C. (2019). Atypical functional connectome hierarchy in autism. *Nature Communications*, 10(1), 1022. <https://doi.org/10.1038/s41467-019-08944-1>
- Jayaprakash, H. J., Mizuno, A., Snitz, B. E., Cohen, A. D., Klunk, W. E., Aizenstein, H. J., & Karim, H. T. (2024). *Voxel-wise hemispheric Amyloid Asymmetry and its association with cerebral metabolism and grey matter density in cognitively normal older adults* (p.

2024.03.05.24303808). medRxiv. <https://doi.org/10.1101/2024.03.05.24303808>

- Karolis, V. R., Corbetta, M., & Thiebaut de Schotten, M. (2019). The architecture of functional lateralisation and its relationship to callosal connectivity in the human brain. *Nature Communications*, *10*(1), 1417. <https://doi.org/10.1038/s41467-019-09344-1>
- Kharabian Masouleh, S., Plachti, A., Hoffstaedter, F., Eickhoff, S., & Genon, S. (2020). Characterizing the gradients of structural covariance in the human hippocampus. *NeuroImage*, *218*, 116972. <https://doi.org/10.1016/j.neuroimage.2020.116972>
- Kong, R., Tan, Y. R., Wulan, N., Ooi, L. Q. R., Farahibozorg, S.-R., Harrison, S., Bijsterbosch, J. D., Bernhardt, B. C., Eickhoff, S., & Yeo, B. T. T. (2023). Comparison between gradients and parcellations for functional connectivity prediction of behavior. *NeuroImage*, *273*, 120044. <https://doi.org/10.1016/j.neuroimage.2023.120044>
- Labache, L., Ge, T., Yeo, B. T. T., & Holmes, A. J. (2023). Language network lateralization is reflected throughout the macroscale functional organization of cortex. *Nature Communications*, *14*(1), Article 1. <https://doi.org/10.1038/s41467-023-39131-y>
- Levy, J. (1977). The Mammalian Brain and the Adaptive Advantage of Cerebral Asymmetry. *Annals of the New York Academy of Sciences*, *299*(1), 264–272. <https://doi.org/10.1111/j.1749-6632.1977.tb41913.x>
- Liang, X., Zhao, C., Jin, X., Jiang, Y., Yang, L., Chen, Y., & Gong, G. (2021). Sex-related human brain asymmetry in hemispheric functional gradients. *NeuroImage*, *229*, 117761. <https://doi.org/10.1016/j.neuroimage.2021.117761>
- Margulies, D. S., Ghosh, S. S., Goulas, A., Falkiewicz, M., Huntenburg, J. M., Langs, G., Bezgin, G., Eickhoff, S. B., Castellanos, F. X., Petrides, M., Jefferies, E., & Smallwood, J. (2016). Situating the default-mode network along a principal gradient of macroscale cortical organization. *Proceedings of the National Academy of Sciences*, *113*(44), 12574–12579. <https://doi.org/10.1073/pnas.1608282113>

- Pang, J. C., Aquino, K. M., Oldehinkel, M., Robinson, P. A., Fulcher, B. D., Breakspear, M., & Fornito, A. (2023). Geometric constraints on human brain function. *Nature*, *618*(7965), 566–574. <https://doi.org/10.1038/s41586-023-06098-1>
- Pilli, V. K., Jeong, J.-W., Konka, P., Kumar, A., Chugani, H. T., & Juhász, C. (2019). Objective PET study of glucose metabolism asymmetries in children with epilepsy: Implications for normal brain development. *Human Brain Mapping*, *40*(1), 53–64. <https://doi.org/10.1002/hbm.24354>
- Vos de Wael, R., Benkarim, O., Paquola, C., Lariviere, S., Royer, J., Tavakol, S., Xu, T., Hong, S.-J., Langs, G., Valk, S., Misic, B., Milham, M., Margulies, D., Smallwood, J., & Bernhardt, B. C. (2020). BrainSpace: A toolbox for the analysis of macroscale gradients in neuroimaging and connectomics datasets. *Communications Biology*, *3*(1), 1–10. <https://doi.org/10.1038/s42003-020-0794-7>
- Wan, B., Bayrak, Ş., Xu, T., Schaare, H. L., Bethlehem, R. A., Bernhardt, B. C., & Valk, S. L. (2022). Heritability and cross-species comparisons of human cortical functional organization asymmetry. *eLife*, *11*, e77215. <https://doi.org/10.7554/eLife.77215>
- Wan, B., Hong, S.-J., Bethlehem, R. A. I., Floris, D. L., Bernhardt, B. C., & Valk, S. L. (2023). Diverging asymmetry of intrinsic functional organization in autism. *Molecular Psychiatry*, 1–11. <https://doi.org/10.1038/s41380-023-02220-x>
- Xu, T., Nenning, K.-H., Schwartz, E., Hong, S.-J., Vogelstein, J. T., Goulas, A., Fair, D. A., Schroeder, C. E., Margulies, D. S., Smallwood, J., Milham, M. P., & Langs, G. (2020). Cross-species functional alignment reveals evolutionary hierarchy within the connectome. *NeuroImage*, *223*, 117346. <https://doi.org/10.1016/j.neuroimage.2020.117346>

Reviewer #1 (Remarks to the Author):

Whereas manuscript has improved, particularly, in explanation of methods and results, I have still considerable concerns about the unsound or redundant application of statistics. Among this: 1) Unsound analytical procedure which the authors called “model explanations and regularisation”; 2) The of generalisation of model parameters to the training set in hemispheric analysis is blown out of proportion to support statements about “hemispheric competition/lateralisation”, “shared optimisation”. 3) confusingly redundant procedure which the authors call the stepwise regression.

We thank the Reviewer for the comments. We have addressed the questions and concerns in the detailed comments below and revised accordingly.

Q1:

Here is a more detailed criticism:

Results

“There was no intercorrelation among these gradients (Supplementary Fig. S2), indicating the statistical independence of gradients.” – Should be a trivial observation

A1: Indeed, this is a trivial observation and for clarity we have now removed this statement.

Q2:

“whether x amount of variance explained in FC by the first y gradients corresponds to x variance explained in the CMRglc map”- I think this needs more explanation, particularly, what is meant by explaining variance in FC in the current context (I hope this is not derived from eigenvalues of diffusion maps, as they do not relate to variance). Furthermore, given that gradients are computed from FC matrix and independently from CMRglc, I am not sure how valid this analysis is. With this in mind, I don’t think I understand what sort of analyses are reported in this paragraph and Fig 3A. “70% of the variance in CMRglc but only 14% in FC, a fivefold difference” - pertaining to the above, this is numerically suspicious. The gradient matrix, presumably, has to be 360 parcels by 100 gradients. (Or not?). Which means the rank of the matrix is > 25% of the number of parcels. How accounted-for variance can be lower than that is not transparent to me. “Validation using null models” this seems to be a nice result, though – as mentioned above - the procedure that led to selection of sparsity parameter was not clear to me and hence As a side note – isn’t application of sparsity and $\alpha=.5$ for diffusion kernel amounts to double (implicit and explicit) attenuation of

non-linearity of embedding? (Which is of relevance, for instance, for the accuracy of Procrustes procedure, see below)? “Finally, we .. “ – “We also” would be more fitting.

A2:

Thank you for the detailed comment and explanation of the results. You are right, we used eigenvalues before and we have now replaced it with the variance explained ratios: eigenvalue for each gradient/sum(eigenvalues). Please see our updated Fig. 3 below. Of note can be better described as the ‘x’ largest diffusion components. We have now further explained this in the results, and termed the previous ‘variance explained’ with ‘cumulative eigenvalue ratio’ to avoid confusion. The conclusion remains the same.

Sparsity is employed on the FC matrix, i.e., top xx % of the connectivity is used to calculate the affinity matrix. The diffusion kernel of $\alpha=0.5$ is then employed on the affinity matrix, and provides density cleaning in non-uniform sampling. So sparsity and alpha are not in parallel (doubled) employed on the FC matrix, $FC \gg \text{sparse FC} \gg \text{affinity} \gg \text{diffusion}$ ($\alpha = 0.5$). $\alpha=0.5$ has been suggested by a lot of previous studies (Hong et al., 2019, 2020; Margulies et al., 2016; Vos de Wael et al., 2020), which balances local and global geometry and considers the density correction. Therefore, we used this number on the affinity matrix (Coifman et al., 2005). We also performed different sparsity options when calculating the affinity matrix, as displayed in Figure 2 and Figure 3. Sparsity=0.9 is a default practice to threshold the weak connections to obtain the reproducible gradients (Hong et al., 2020; Vos de Wael et al., 2020). Our own additional reasoning to use sparsity=0.9 to do following analyses after Figure 3A was that gradients explained to CMRglc map (e.g. 50%) with sparsity=0.9 using less accumulated variance ratio. We revised the text:

“Next, to determine how dense (i.e., sparsity) the FC can show a best CMRglc-gradients reconstruction, we plotted the CMRglc explanation by gradients (y-axis) and accumulated variance ratio of FC gradients (x-axis) in Fig. 3A. Even though different sparsities of FC can reach >70% finally CMRglc reconstruction, sparsity = 0.9 reached faster than other numbers. For example, we found under the 50% of cumulative eigenvalue ratios, the gradients from sparsity = 0.9 can reach around 35.7% variance of CMRglc and other sparsities were around 20%. The performance of sparsity = 0.9 was above other numbers. This difference suggests that weak connectivity may be less useful to reconstruct metabolic structure.”

A. Gradients explanation

B. Feature importance using 100 gradients

“Fig. 3: Model explanation and regularization. A). plots the scatter with the x-axis showing gradient variance ratios (each gradient eigenvalues by total eigenvalues) and the y-axis showing CMRglc variance explained by gradients. The lower panel further displays how their relationship (slope) changes by model with more gradients entered (starting from 10). B). plots scatter using elastic net regularization (lasso/ridge = 1) onto the step 100 model: one without penalty (standardized regression coefficient and 95% confidence interval by number of gradients), and the other with six penalty parameters (α) from 0.05 to 0.3 (slope by number of gradients). Pearson correlation coefficients (r) were calculated using true CMRglc and model-predicted CMRglc maps.”

Q3:

“The CMRglc asymmetry map was calculated by z-scoring the LH and RH separately” – unclear motivation for separate z-scoring.

A3:

Thank you for noticing this. We aimed to correct for the global effects resulting in left-right differences when performing regional asymmetry analysis. Z-scoring for different hemispheres can help avoid that one hemisphere plays a global stronger role. This step improves comparability between hemispheres. However, for the whole brain modeling, the z-scoring was for the whole brain and only asymmetry part we used z-score of different hemispheres. And the correlation between z-scoring the whole brain and z-scoring separate

hemispheres is $r = 0.9999$ (whole brain map) and $r = 0.999$ (asymmetry map). We also performed the analysis with z-scoring the whole brain, the float of results changes from the fourth decimal place, where we only keep three decimal places in our results.

Q4:

As for the previous version of the manuscript, I do not understand the logic and find it extremely convoluted. I believe Fig 5C shows that FC gradients in one hemisphere accounts similarly for CMRglc in both hemispheres, with FC gradients in RH making a somewhat poorer predictions for higher number of gradients. How “energy optimization”, “energy sharing”, “shared optimization”, “lateralisation” and “competition” follow from this is beyond my comprehension. E.g., “hemispheric competition” typically refers to a dynamic process, how this can be reflected in the association between two static features? There is no “sharing” of energy here, only generalisable modelling parameters. Who shares? Gradients? What is the logical link between “hemispheric lateralisation” (typically meaning that something is present in one hemisphere but less so in the other) and the fact that the one hemisphere gradients account for energy in both hemispheres? What “optimisation” can even possibly mean here? I feel that all this terminology is just a cover-up for a very simple, perhaps an uninspiring, result that the interhemispheric difference in energy maps are not great enough to affect generalisation of coefficients across hemispheres (which might be an inadvertent consequence of z-scoring two hemispheres separately)

No explanation of Fig 5E in the text.

A4:

We thank the Reviewer for this detailed critique, which has helped us recognize that our original framing was indeed overly complex and used terminology in an unclear manner. We have substantially revised this section to address these concerns.

Major changes:

1. Removed misleading terminology and related contents in abstract. We have eliminated all references to "energy optimization," "competition," "sharing," and "lateralization" in this context. The reviewer is correct that these terms imply dynamic processes or theoretical frameworks that are not supported by our cross-sectional analysis of static features. We now use precise, descriptive language: "hemisphere-specific models" vs. "cross-hemisphere generalization models."
2. Simplified the conceptual framework. Our analysis addresses a straightforward question: Is the relationship between functional gradients and glucose metabolism similar across hemispheres (allowing cross-hemisphere prediction) or

- hemisphere-specific (requiring separate models)? We now present this clearly without unnecessary theoretical overlay.
3. Acknowledged limitations in model performance. The reviewer's skepticism was warranted. Our analysis reveals that:
 - Even hemisphere-specific models show modest predictive performance (slopes < 0.60)
 - Cross-hemisphere models fail dramatically (slopes ~0.07-0.11), predicting near-constant values
 - These findings could reflect genuine hemispheric differences, methodological limitations in spatial alignment across hemispheres, or both
 4. Addressed the z-scoring concern. We tested whether separate hemisphere z-scoring affected our results. The correlation between separate and whole-brain z-scoring approaches was $r=0.999$, indicating that our normalization procedure removes similar information regardless of approach. This actually supports the Reviewer's point that absolute hemispheric differences are being normalized away, and we acknowledge this limitation in the revised text.
 5. Toned down conclusions. Rather than claiming evidence for "optimization" or "competition," we now state that hemisphere-specific models perform numerically better than cross-hemisphere models, but that the overall weak predictive performance and potential methodological confounds (spatial alignment, normalization) limit strong biological conclusions.
 6. Added explanation of Fig 5E as requested (see revised text below).

The Reviewer is correct that our original interpretation overreached. The revised version presents our findings more accurately as exploratory evidence that energy-function relationships may have some hemispheric specificity, while acknowledging that methodological factors could contribute to this pattern.

Results:

“Next, we investigated whether the relationship between functional organization and glucose metabolism is consistent across hemispheres or shows hemispheric specificity. We calculated functional gradients in the left and right hemispheres separately, with right hemisphere (RH) gradients aligned to left hemisphere (LH) gradients using Procrustes rotation (Fig. 5A). We then calculated asymmetry maps (LH-RH) for both functional gradients and CMRglc, with CMRglc values z-scored within each hemisphere (Fig. 5B).

We trained models on the LH data and then used the trained parameters to predict CMRglc in the RH and vice versa for another hemisphere (Fig. 5C). We then defined five model

variants to compare hemisphere-specific versus cross-hemisphere prediction (Fig. 5D-E): Mode 1 (LH-RH specific): train and predict within own hemisphere; Mode 2 (RH-LH specific): train and predict with exchange of hemisphere; Modes 3-4 (generalization models): train on one hemisphere predict the both; Mode 5 (asymmetry-based): train on gradient asymmetry to predict CMRglc asymmetry. We evaluated predictions using covariance and slope metrics (Fig. 5F).

Hemisphere-specific models showed modest performance (mode 1: covariance=0.058, slope=0.603; mode 2: covariance=0.041, slope=0.424), while cross-hemisphere models performed substantially worse (modes 3-4: covariance<0.011, slopes<0.12), with slopes indicating near-constant predictions. The asymmetry-based model showed intermediate performance (covariance=0.051, slope=0.530). Predicted maps are shown in Supplementary Fig. S4. We could also replicate the finding using joint alignment between left and right hemispheres (Supplementary Fig. S5), which was qualitatively similar to Procrustes alignment. In particular, we found the variance for the modes 1 and 2 was 0.059 and 0.082, but for the modes 3, 4 and 5 was 0.019, 0.043, 0.053. The slopes for the modes 1-5 were: 0.616, 0.856, 0.208, 0.448, and 0.550.

Overall, these results suggest some degree of hemispheric specificity in energy-function relationships. However, several limitations warrant caution. First, even the best models explain only half of metabolic asymmetry (slope=0.530). Second, the less fitting of cross-hemisphere generalization may partly reflect methodological challenges in spatial alignment across hemispheres. Third, our z-scoring procedure normalizes away absolute hemispheric differences, focusing on relative spatial patterns within each hemisphere. We therefore interpret these findings as preliminary evidence for hemispheric specificity, acknowledging that weak predictive performance and potential methodological confounds limit strong biological conclusions.”

Discussion:

“By comparing hemispheric-generalization and -specific models we found that energy asymmetry was better explained when the hemispheres were treated as separate systems, yet overall comparative effect was weak.”

“Brain lateralization has been proposed as a mechanism for optimizing brain energy requirements²². Here we evaluated this hypothesis by using asymmetry maps of functional organization to model the energy asymmetry map. We found that hemispheric specific models outperformed the generalization models with a small winning rate of 14%. This suggests that energy asymmetry may not result solely from cumulative functional lateralization (generalization models) in topology. While asymmetry of functional gradients has been

extensively studied^{15,17,18,20,21}, our results suggest that gradients may not only reflect static brain organization but also how the brain operates across different spatial manifolds in the form of independent hemispheric optimization in the context of competing for energy source. Such an interpretation could be in line with conceptualizations that functional lateralization results from interactions between hemispheric functions^{55,56}. For example, hemispheric competition describing the changing role of hemispheres has been observed during attention and sleep over time⁵⁷⁻⁶⁰. Our findings give a novel insight that energy lateralization may be the result of such functional competition across spatial manifolds in addition to temporal scales.”

Q5:

Methods

“This procedure optimally rotates, translates, and scales the right hemisphere” – this doesn’t account though for non-linearity of Laplacian embedding which means that distances across different embeddings are not by default isomorphic. A diagnostic of spectral gaps’ similarity across embedding dimensions can be advised to ascertain there are no considerable non-linear distortions for which Procrustes can’t compensate. Otherwise, a multi-view embedding is warranted.

A5:

Thanks for this point. Now we first calculated the similarities for the left and right dimensions using Procrustes alignment: range from 0.563 to 0.983 with mean: 0.823 and standard deviation: 0.109. Then, we performed the analysis using a joint alignment approach to provide the multi-view embedding. Interestingly, joint alignment produced qualitatively similar results to Procrustes alignment (Supplementary Fig. S5). Hemisphere-specific models still outperformed cross-hemisphere generalization (mode 1: covariance=0.059, slope=0.616; mode 2: covariance=0.082, slope=0.856 vs. modes 3-4: covariance=0.019-0.043, slopes=0.208-0.448). In fact, cross-hemisphere generalization performed slightly worse with joint alignment, further supporting that hemispheric specificity is not simply an artifact of Procrustes misalignment. We showed the Supplementary Figure below. The results have been added in above Q4.

A. Cross-hemispheric generalization

B. Mode comparisons

Figure S5. Replication analysis using joint alignment between left and right hemispheres. **A.** The cross-hemispheric generalization corresponding to **Figure 5C**. We used gradients after joint alignment to perform this analysis. **B.** Mode comparisons corresponding to **Figure 5F**. It compares fitting covariance between the specific (modes 1 and 2, pink curves) and generalization (modes 3, 4, and 5, grey curves) models.

We revised the Methods:

“In addition, to provide a multi-view of the alignment, we used joint alignment between LH and RH for replication. This approach computes LH and RH gradients simultaneously in a shared embedding space using concatenated LH and RH connectivity matrices^{36,61}.”

Q6:

“to generate 100 principal gradients for each individual and at the group level” - there is some ambiguity what “group level” means in this context. Embedding of average affinity matrix? Perhaps worth giving a hint of what will be done with individual embeddings in downstream analyses.

A6:

We have updated the statement: *“Group-level functional organization gradients were computed by reducing the dimensionality of the affinity matrix of the mean functional*

connectome across individuals using diffusion map embedding.” We hope it is now clearer that the group-level gradient is calculated from the group averaged FC matrix. Individual gradients are calculated from individual FC matrices.

Q7:

“The name of dimensionality reduction, which belongs to the family of graph Laplacians,” – I don’t think the name was stated.

A7:

The name is diffusion maps. We have revised the sentence: “*The name of this diffusion-embedded mapping, which belongs to the family of graph Laplacians, is derived from the equivalence of the Euclidean distance between points*”.

Q8:

“The stepwise regression was done using “ – I took a note of the author’s response, but this is just off the point, except that I now understood what exactly was done using “step-wise” regression. The step-wise regression is almost a misnomer here, at least in a conventional way. Unlike the cases where conventional step-wise regression is typically applied, the gradients are orthogonal by construction, which means the whole procedure can be accomplished by fitting one – 100-gradient - model, and computing unique variance accounted for by each gradient and summing them up as needed. What the authors do is not *wrong*, the result will be equivalent, but the manner in which the result is achieved is redundant/cumbersome and may confuse the reader.

A8:

Thanks for this comment. To clarify our analysis further, we have now replaced “stepwise regression” with reconstruction, as the (Pang et al., 2023) indicated. Regarding “the manner in which the result is achieved”, we emphasize more on the importance of the gradient order, which makes more sense, story-wise, to exhibit the Figure 3.

Q9:

“Adjusted R-squared decreases or is negative when adding more variables into the model in general, indicating collinearity of the model and the need to adjust the entered variables. In the current analysis, we did not find collinearity between gradients” – this is very confusing,

mixing apple and oranges. They may decrease even if the predictors are not collinear; for orthogonal variables, this would just signify that a particular variable explains nothing in the data.

A9:

We thank the Reviewer for this clarification. The Reviewer is correct that we conflated two distinct issues: collinearity between predictors and lack of predictive power for the outcome.

We have revised the text to clarify: "*Adjusted R-squared penalizes model complexity and can decrease when adding variables that contribute little predictive power, even when predictors are orthogonal to each other. While gradients are orthogonal by construction in the embedding space, a decrease in adjusted R-squared when adding higher-order gradients indicates that these gradients have weak relationships with glucose metabolism and do not improve prediction.*"

This correctly reflects that the decreases in adjusted R-squared we observe are due to higher-order gradients explaining little variance in CMRglc, not due to collinearity among gradients.

References

- Coifman, R. R., Lafon, S., Lee, A. B., Maggioni, M., Nadler, B., Warner, F., & Zucker, S. W. (2005). Geometric diffusions as a tool for harmonic analysis and structure definition of data: Diffusion maps. *Proceedings of the National Academy of Sciences*, *102*(21), 7426–7431. <https://doi.org/10.1073/pnas.0500334102>
- Hong, S.-J., Vos de Wael, R., Bethlehem, R. A. I., Lariviere, S., Paquola, C., Valk, S. L., Milham, M. P., Di Martino, A., Margulies, D. S., Smallwood, J., & Bernhardt, B. C. (2019). Atypical functional connectome hierarchy in autism. *Nature Communications*, *10*(1), 1022. <https://doi.org/10.1038/s41467-019-08944-1>
- Hong, S.-J., Xu, T., Nikolaidis, A., Smallwood, J., Margulies, D. S., Bernhardt, B., Vogelstein, J., & Milham, M. P. (2020). Toward a connectivity gradient-based framework for reproducible biomarker discovery. *NeuroImage*, *223*, 117322. <https://doi.org/10.1016/j.neuroimage.2020.117322>
- Margulies, D. S., Ghosh, S. S., Goulas, A., Falkiewicz, M., Huntenburg, J. M., Langs, G., Bezgin, G., Eickhoff, S. B., Castellanos, F. X., Petrides, M., Jefferies, E., & Smallwood, J. (2016). Situating the default-mode network along a principal gradient of macroscale cortical organization. *Proceedings of the National Academy of Sciences*, *113*(44), 12574–12579. <https://doi.org/10.1073/pnas.1608282113>
- Pang, J. C., Aquino, K. M., Oldehinkel, M., Robinson, P. A., Fulcher, B. D., Breakspear, M., & Fornito, A. (2023). Geometric constraints on human brain function. *Nature*, *618*(7965), 566–574. <https://doi.org/10.1038/s41586-023-06098-1>

Vos de Wael, R., Benkarim, O., Paquola, C., Lariviere, S., Royer, J., Tavakol, S., Xu, T., Hong, S.-J., Langs, G., Valk, S., Misic, B., Milham, M., Margulies, D., Smallwood, J., & Bernhardt, B. C. (2020). BrainSpace: A toolbox for the analysis of macroscale gradients in neuroimaging and connectomics datasets. *Communications Biology*, 3(1), 1–10. <https://doi.org/10.1038/s42003-020-0794-7>